



***Title:*** *Nordic Seas Deep-Water susceptible to enhanced freshwater export to the subpolar*
*North Atlantic during peak MIS 11*
Michelle J. Curran [a], Christophe Colin [b], Megan Murphy O'Connor [a,c], Ulysses Ninnemann [d], and
Audrey Morley [a,c,*]
*[a] School of Geography, Archaeology and Irish Studies, and The Ryan Institute at the University of Galway,*
*H91TK33 Galway, Ireland.*
*[b] Université Paris-Saclay, CNRS, GEOPS, 91405, Orsay, France*
*[c] iCRAG – Irish Centre for Research in Applied Geosciences, Belfield, Dublin 4, Ireland*
*[d] Department of Earth Science and Bjerknes Centre for Climate Research, University of Bergen, Bergen, Norway*
*\* Corresponding Author*

## 15 Abstract

Recent investigations into Marine Isotope Stage (MIS) 11 (424-403 ka), an unusually long and warm
interglacial of the Quaternary Period, have found that the Atlantic Meridional Overturning Circulation
remained strong while background melting of the Greenland Ice-Sheet (GIS) was high, and resulted in
a fresh and cold surface ocean in the Nordic Seas. These investigations support the hypothesis that
deep-water formation may not be as susceptible to future GIS melting as previously thought. Here we
test this hypothesis and present a palaeoceanographic investigation of a freshwater-related abrupt
climate event recorded in the eastern North Atlantic during peak interglacial conditions (~412 ka),
when the GIS was as small or smaller than today. Using sediment core DSDP-610B recovered from the
western Rockall Trough we reconstruct the evolution of Nordic Seas Deep-Water (NSDW) using
benthic carbon isotope, Neodymium isotopes, and grain-size analysis paired with end-member
modelling. Further, a combination of planktonic foraminiferal assemblage census and Ice-Rafted
Debris counts allow us to reconstruct surface water properties including temperature and the
movement of oceanic fronts throughout this event. Our results demonstrate that a reduction of NSDW
only occurs once GIS melt and polar freshwater reaches subpolar latitudes. We hypothesise that the
reorganisation of fresh and cold surface waters from the Nordic Seas into the subpolar North Atlantic
was responsible for an AMOC-related cold event centred at 412 ka. Placing our results in the
palaeogeographical context of the North Atlantic Region we tentatively propose that the ocean-
atmosphere climate dynamics linking the Nordic Seas with the subpolar North Atlantic played and will
play a crucial role for the stability of NSDW formation in the future, considering the enhanced melting
and overall hydrological cycle at high Northern latitudes predicted for future climate scenarios.

## 37 1 Introduction



Modern Greenland Ice-Sheet (GIS) melting is a response to increased global mean temperatures
driven by rising greenhouse gas emissions (Aguiar et al., 2021;Fettweis et al., 2017;Tedesco and
Fettweis, 2012;Golledge et al., 2019). The addition of meltwater has the potential to alter surface
water buoyancy (Østerhus et al., 2001;Praetorius et al., 2008) and thereby deep-water formation
(Galaasen et al., 2014;Bond et al., 1997) at high-latitudes. This is pertinent for future climate change
scenarios, as multiple modelling studies suggest that the strength of the Atlantic Meridional
Overturning Circulation (AMOC) may be impacted by meltwater (Stommel, 1961;Rahmstorf,
1995;Caesar et al., 2018), derived from the GIS (Bakker et al., 2016;Böning et al., 2016;Luo et al.,
2016;Yu et al., 2016). Modern observations over the past 30 years, however, do not confirm a decline
or sustained weakening of the AMOC in response to increased freshwater export to deepwater
formation regions (Worthington et al., 2021), highlighting the need for longer observations reaching
beyond instrumental datasets.

Past archives of North Atlantic Deep Water (NADW) variability provide us with a tool to assess NADW
stability and response to changing climate boundary conditions. This is important as recent palaeo
studies by Caesar et al. (2021) and Thornalley et al. (2018) suggest the AMOC is currently at its weakest
state for the past ~150 years. Thornalley et al. (2018) link the AMOC slowdown to freshwater runoff
from the GIS. Therefore, it is crucial to improve our understanding of how surface and deep-water
components within the North Atlantic are linked, and on what time frames they respond to
freshwater-induced climate instabilities during interglacial climate boundary conditions. Past
interglacials provide a good analogue to enhance our understanding of the complex nature of the
climate system (Yin and Berger, 2015). In particular, periods with sufficiently similar boundary
conditions to the present day can meaningfully advance our understanding of the antecedents and
mechanisms generating variability and instability in the climate system.

Marine Isotope Stage (MIS) 11 began at ~424 ka (Lisiecki and Raymo, 2005) and was a particularly
long, ~30 ka, interglacial (Howard, 1997;McManus et al., 1999;Reyes et al., 2014), covering two
precessional cycles (Laskar et al., 2004). Unlike the current Holocene, the manifestation of the climate
optimum occurred relatively late into the interglacial, i.e., after ~410 ka (Ruddiman, 2005;Tzedakis et
al., 2012;McManus et al., 1999;Kandiano and Bauch, 2007;Dickson et al., 2009), most likely due to the
anti-phasing of precession and obliquity (Laskar et al., 2004). Nevertheless, MIS 11 is often cited as a
good analogue for our current interglacial (Tzedakis et al., 2012;Droxler et al., 2003;Berger and Loutre,
2002;Loutre and Berger, 2003;Candy et al., 2014;Tzedakis, 2010;Mcmanus et al., 2003) due to
persistently high atmospheric $CO_2$ concentrations (Petit et al., 1999;Raynaud et al., 2005;Nehrbass-



Ahles et al., 2020), similar to preindustrial Holocene values (Bazin et al., 2013) (Figure 1), and
dampened precession modulated by an eccentricity minimum (Berger and Loutre, 1991;Bauch et al.,
2000;Hodell et al., 2000;Dickson et al., 2009;Loutre and Berger, 2000;Bazin et al., 2013).

The peaks in precession at 424 ka and 408 ka (Pol et al., 2011;Tzedakis, 2010), contributed to
protracted high-latitude warming. Paired with high $CO_2$ concentrations these boundary conditions
resulted in excessive GIS melt/retreat culminating in minimum GIS extent at ~403 ka (Robinson et al.,
2017a). Due to the retreat of high-latitude ice sheets, MIS 11 had relative sea levels ~6 to 13 metres
above present-day values, peaking at ~403 ka (PAGES, 2016;Reyes et al., 2014;Robinson et al.,
2017a;Raymo and Mitrovica, 2012). Further, terrestrial archives provide evidence for sustained
warming of the North Atlantic and Arctic regions during MIS 11 (Melles et al., 2012;Desprat et al.,
2005a;Nitychoruk et al., 2005;Prokopenko et al., 2010;SHICHI et al., 2009;Ashton et al., 2008;Preece
et al., 2007;Reille et al., 2000;Tzedakis et al., 1997;Tzedakis et al., 2006) demonstrating warmer than
present boundary conditions at least at high Northern latitudes. However, several palaeoceanographic
studies document lower than present Sea Surface Temperature (SST) in the Nordic Seas (Kandiano et
al., 2016;Helmke and Bauch, 2003;Bauch et al., 2000;Doherty and Thibodeau, 2018;Thibodeau et al.,
2017;Kandiano et al., 2012).

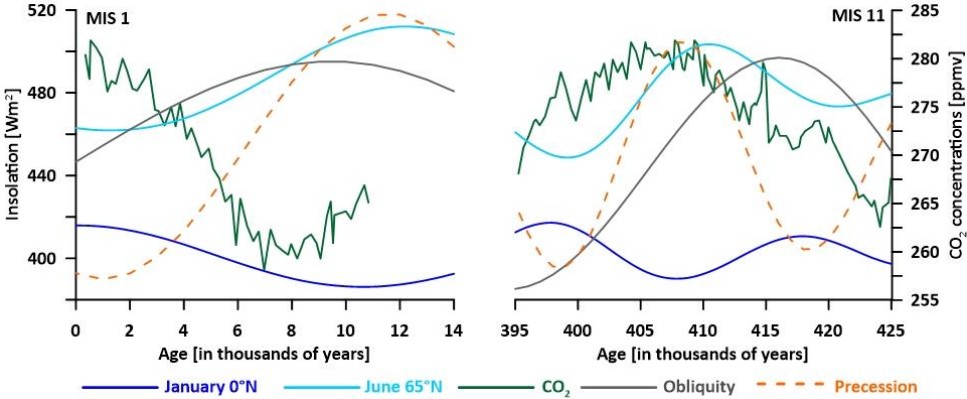

Figure 1: Climate forcings during MIS 1 and MIS 11. January insolation at 0°N [dark blue], June insolation at 65°N
[light blue], Obliquity [grey], and Precession [dashed orange] (Laskar et al., 2004). Antarctica $CO_2$ (ppmv) [green]
– MIS 1 (Bazin et al., 2013) – MIS 11 (Nehrbass-Ahles et al., 2020).

The leading hypothesis explaining these low SSTs postulates that the prolonged high-latitude warming
enhanced freshwater export via melting from adjacent ice-sheets (Thibodeau et al., 2017;Kandiano et
al., 2017a) and/or amplified Eurasian river runoff (Doherty and Thibodeau, 2018) throughout MIS 11
(de Vernal and Hillaire-Marcel, 2008;PAGES, 2016;Reyes et al., 2014). Increased freshwater at high
latitudes then likely resulted in cool and relatively fresh buoyant surface water (Kandiano et al.,



2017a;Thibodeau et al., 2017;Kandiano et al., 2012) at least until ~411 ka (Kandiano et al., 2012). Yet,
despite the presence of freshwater, Nordic Sea Deep Water (NSDW) formation is believed to have
been generally vigorous throughout MIS 11 (Dickson et al., 2009;Riveiros et al., 2013;McManus et al.,
1999) although it's short-term stability/variability has been questioned (Galaasen et al. 2020). This
scenario is contrary to models (Brodeau and Koenigk, 2016;Stouffer et al., 2006) and palaeo-
observations of the recent past (Caesar et al., 2021;Thornalley et al., 2018).
An abrupt climate event at ~412 ka on past thresholds for reorganizing the climate-circulation regime
as the boundary conditions (e.g. freshwater fluxes and distribution) evolve within the Nordic Seas and
North Atlantic Basin. This event is recorded as a sea surface cold event (Kandiano et al., 2017b;Barker
et al., 2015;Irvalı et al., 2020;McManus et al., 1999;Alonso-Garcia et al., 2011) across the subpolar
North Atlantic. A concurrent perturbation of stable carbon isotopes ($\delta^{13}$C) measured on benthic
foraminifera (Galaasen et al., 2020;Hodell et al., 2008;McManus et al., 1999;Riveiros et al., 2013)
further suggests a connection with the deep ocean. Continuous background melting or an abrupt
meltwater discharge from the GIS, have been proposed as a trigger for this event, similar to events
identified during other interglacials of the Quaternary (Galaasen et al., 2014;Galaasen et al.,
2020;Irvalı et al., 2020;Irvalı et al., 2016). However, the mechanisms and phase relationships linking
the release of freshwater across the Subpolar Gyre (SPG) to a slowdown in NSDW, remain elusive due
to the low temporal resolution of existing records.
Here we present a detailed investigation of the climate-ocean perturbation at 412-ka, focusing on the
temporal evolution (leads/lags) between surface and deep-water changes, within the eastern North
Atlantic. We reconstruct both SST and deep-water properties from the same samples of Deep Sea
Drilling Project (DSDP) site 94-610B (610B). Site 610B lies in the path of both the surface North Atlantic
Current (NAC) and deep-water Wyville-Thomson Overflow Water (WTOW), a conduit of NSDW. This
unique location and approach allow us to assess the relative timing between climate forcing and the
response in the surface and deep branch of the AMOC located in the eastern North Atlantic. We are
thus able to test the hypothesis linking background melting to a weakening of NSDW formation.
Specifically, we aim to improve our mechanistic understanding of the climate response to meltwater
forcing – a mechanism likely pertinent to the future evolution of the ocean-atmosphere climate
system.
**2 Hydrographic setting and materials**
Modern sites for deep-water formation in the North Atlantic are the Nordic Seas and the subpolar
North Atlantic, including the Irminger and Labrador Seas (Sgubin et al., 2017a). Overturning in the
Nordic Seas is primarily modulated by thermohaline forcing (Hansen and Østerhus, 2000). The



production of Iceland Scotland Overflow Waters (ISOW) and the Denmark Strait Overflow Waters
(DSOW) creates density and pressure gradients at depths that drive overflow transport (Hansen and
Østerhus, 2000;Olsen et al., 2008). As a result the outflow at depth creates a pressure gradient at the
surface between the North Atlantic and the Nordic Seas (Jungclaus et al., 2006a;Mauritzen,
1996;Doherty et al., 2021;Olsen et al., 2008;Østerhus et al., 2001;Hansen and Østerhus, 2000) forcing
a compensating inflow of Atlantic Waters into the Nordic Seas (Østerhus et al., 2001;Hansen and
Østerhus, 2000). Thus, deep-water formation in the Nordic Seas (NSDW) may continue despite
enhanced freshening, as long as a density gradient across the Greenland-Scotland Ridge, connecting
the Atlantic and the Nordic Seas, is maintained (Østerhus et al., 2001) and the inflow of Atlantic Water
via wind stress continues (Sandø et al., 2012). Overturning in the Irminger and Labrador Seas mainly
occurs in winter as a result of buoyancy loss to the atmosphere (Sgubin et al., 2017a;Brodeau and
Koenigk, 2016). As a result these sites likely respond more rapidly to freshwater input (Latif et al.,
2006), which could stabilise the water column and thereby weaken overturning, as modelled by
Jungclaus et al. (2006a) and Olsen et al. (2008). To assess the full impact of freshwater on the
variability and strength of deep-water formation, long-term observations are required because of the
decadal to multidecadal integration of surface water variability in the deep ocean (Buckley and
Marshall, 2016). However, in-situ observations are only available since 2004 (Smeed et al., 2018)
leaving us with a gap in our understanding of how NADW responds to sustained freshwater forcing.

Site 610B was recovered from the Feni Drift in the Rockall Trough, in the eastern North Atlantic at
53°13.297'N, 18°53.213'W; 2417 metres below sea-level (m) (Figure 2). The Feni Drift is a sedimentary
contourite formed by overflow currents between the early Oligocene (33.9-23 Ma) and late Pliocene
(3.6-2.58 Ma), with high accumulation rates on the crest of the Drift during the Pleistocene (Naylor
and Shannon, 2005;Robinson and McCave, 1994;Flood et al., 1979). High-resolution (3.5 kHz) seismic
profiles of the core site show tectonically stable sediment waves that have not migrated since the late
Pliocene (Kidd and Hill, 1987, 1986); therefore we infer that the sedimentary sequence at the core site
remained intact throughout the Quaternary.
At the surface the Rockall Trough directs ~50% of Atlantic Waters into the Nordic Seas (Hansen and
Østerhus, 2000), thereby providing high-salinity Atlantic Waters for NSDW formation. There are two
major sources of surface waters in the Rockall Trough: NAC derived from the Gulf of Mexico (Sutton
and Allen, 1997) and subtropical gyre (STG) derived waters (Hátún et al., 2005). At depth, NSDW enters
the Rockall Trough via Wyville-Thompson Ridge as Wyville-Thompson Overflow Water (WTOW) (Ellett
et al., 1986;Johnson et al., 2017), and accounts for 10-15% of southward flowing NSDW (Dickson and
Brown, 1994; Hansen and Østerhus, 2000). Unlike other water masses in the Rockall Trough, WTOW



is the only water mass that enters from the north and flows south along the western margin of the
Rockall Trough (Johnson et al., 2010). WTOW is limited to the western boundary of the Trough and is
linked with the sedimentary contourite deposits of the Feni Drift (Holliday et al., 2000;Ellett and
Martin, 1973). The southward flow of deep WTOW is intermittent on annual timescales but positive
on decadal timescales (Johnson et al., 2017). To the north and west of site 610B the central
anticyclonic gyre of the Rockall Trough (Johnson, 2012; New and Smythe-Wright, 2001; Smilenova et
al., 2020), recirculates water down to 2000m during winter mixing (Smilenova et al., 2020). Given the
distance from the gyre (ca 500 km) and the deeper depth of site 610B, it is unlikely that this influences
the sedimentation and flow over the site. Modern hydrographic data thus indicates that site 610B lies
in the pathway of poleward flowing Atlantic Waters at the surface and southward flowing WTOW at
depth (Ellett et al., 1986;Johnson et al., 2017). The paired surface and deep-water reconstructions at
site 610B are thus ideal to record both surface and deep-water flow variability in the eastern North
Atlantic.

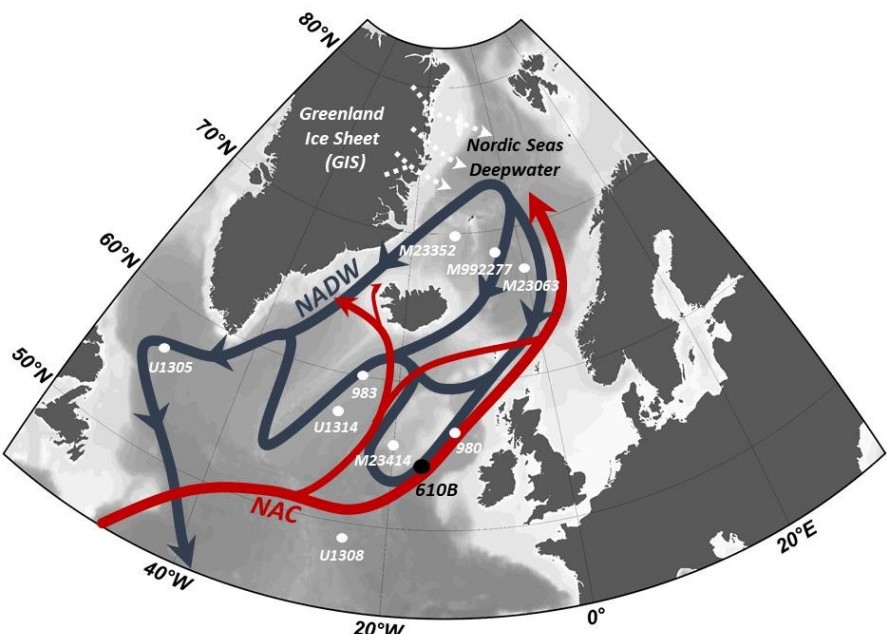


Figure 2: Schematic representation of the North Atlantic Ocean and Nordic Seas with arrows indicating the
circulation components of the Atlantic Meridional Overturning Circulation (AMOC) in the North Atlantic basin.
Major ocean currents include the North Atlantic Current (NAC) in red, deep-water originating in the Nordic Seas
(NSDW) in blue, the Greenland Ice-Sheet (GIS), and freshwater influx routes to the Nordic Seas in white. Also
shown in black are site 610B (this study), site M23414, M992277, M23352, M23063, Ocean Drilling Project (ODP)
sites 980, and 983, and Integrated Ocean Drilling Program (IODP) sites U1314, U1305, and U1308 marked by
white circles. The grey scale signifies bathymetry (GEBCO 2014), and was generated with Ocean Data View
software (http://odv.awi.de/).



### 3 Methods

Here we present data with a sampling resolution of 2.5 cm, between 28.6-29.7 metres below sea-floor (mbsf), and ~4 cm resolution between 29.74-29.83 mbsf, totalling 54 samples. Each sample was split and ~1 g of dry sediment was reserved for grain size analysis when samples had enough sediment available for analysis. The remaining sample was disaggregated on a Stuart SSL1 orbital shaker and washed at >63 μm. Following CLIMAP and Members (1976) selection of foraminiferal specimens for census and Ice-Rafted Debris (IRD) counts were performed after dry sieving to >150 μm.

*3.1 Planktonic foraminiferal counts and species abundance (%)*

Each sample was split using a micro-sample splitter into aliquots containing ~300 planktonic foraminiferal specimens. We identified, counted, and stored each specimen within a sample in identification slides. The absolute number of planktonic foraminifera counted ranged from 300-445. We use the abundance records (e.g., *Neogloboquadrina pachyderma*, (Np), *Turborotalita quinqueloba,* (*T. quinqueloba*)) to reconstruct the advance and retreat of the Polar and Sub-Arctic Fronts (Alonso-Garcia et al., 2011;Mokeddem et al., 2014). Here we define the Polar Front as the boundary between Polar and Arctic Waters, and the Sub-Arctic Front (SAF) as the boundary between Arctic and Atlantic Waters. In the modern ocean Np is the predominant foraminifera north of the Polar Front (Kipp, 1976) and is thus associated with Polar Waters from high-latitudes (Kohfeld et al., 1996;Pflaumann et al., 1996). *T. quinqueloba* is linked to the SAF (Loubere, 1981;Johannessen et al., 1994), with maximum abundance observed on the warmest side of the SAF (Johannessen et al., 1994). The Np coiling ratio is commonly used to infer changes in SST, e.g., Irvalı et al. (2016). We calculate the coiling ratio as Np/(Np + *N. incompta*)*100.

*3.2 Sea Surface Temperature (SST) reconstruction*

We used the ForCenS database (Siccha and Kucera, 2017) and the ROIJA package (Juggins, 2017) in R (Team, 2019), and a squared chord distance (dissimilarity measure), to estimate Modern Analogue Technique (MAT)-derived SSTs (Hutson, 1980;Prell, 1985). Here we chose to reconstruct annual SST from World Ocean Atlas 98, rather than seasonally based SST, because planktonic foraminifera inhabit a wide vertical range within the water column, and exhibit distinct variability in their seasonal abundance (Jonkers et al., 2013). This is particularly persuasive at subpolar latitudes and specifically during interglacial climates when many subpolar species display a double peak in abundance, one in spring and one in late summer (Chapman, 2010). Thus, annual reconstructions are likely more representative of assemblage ecological preferences. A high dissimilarity coefficient indicates poor



modern analogues, with no similar analogue existing in the core-top database. Core tops with
dissimilarity >0.4 were not considered. In this study the average SST standard deviation is 1.6°C.

*3.3 Ice-Rafted Debris (IRD) counts*
The relative abundance of IRD is an established proxy of ice sheet variability (Baumann et al., 1995;
Fronval and Jansen, 1997; Jansen et al., 2000). All grains >150 µm in the aliquots split for census
identification were counted. 10% of samples were recounted to determine the standard error. The
average standard error for IRD counts in this study was 0.9%. We statistically compared the IRD counts
using a t-test (two-sample t-test, p <0.05) and found the differences to be insignificant (*d.f.*=4, *t*=2.78,
*p*=0.2). We present the results as the number of lithogenic/terrigenous grains per gram (grains/g) of
dry sediment.

*3.4 Stable Isotope Analysis*
Stable isotopes of oxygen and carbon were measured from the tests (<150 µm) of benthic foraminifera
*Cibicidoides wuellerstorfi*. In total 60 samples were analysed between 28.55 – 29.955 mbsf. Stable
isotope analyses were measured using a Kiel IV and MAT253 mass spectrometer at FARLAB at the
Department of Earth Science and the Bjerknes Centre for Climate Research, University of Bergen.
Results are expressed as the average of the replicates and reported relative to Vienna Pee Dee
Belemnite (VPDB), calibrated using NBS-19 and crosschecked with NBS-18. Long-term reproducibility
(1σ SD) of in-house standards for samples between 10 and 100 mg is better than 0.08‰ and 0.03‰
for $\delta^{18}O$ and $\delta^{13}C$, respectively.

3.5. *Neodymium isotope measurements on planktic foraminifer*
In total 17 samples of 15 to 30 mg of mixed planktonic foraminifera of size fraction <150 µm were
picked for Nd isotope analysis between 28.6 and 29.935 mbsf of ODP Site 610B. No oxidative-reductive
leaching procedure was employed, and this approach has been demonstrated to be suitable for
extracting bottom water Nd isotopic compositions (Wu et al., 2015). The cleaning procedure and
purification of Nd were carried out in a class 100 clean laboratory using ultrapure reagents. The
foraminifera shells were crushed between two glass slides to open chambers, and the calcite
fragments were ultrasonicated for 1 min in Milli-Q water before pipetting off the suspended particles.
This step was repeated until the water was clear and free of clay particles. Samples were inspected
under a binocular microscope to ensure that all sediment particles had been removed before they
underwent weak acid leaching for 5 min in 1 ml 0.001 M $HNO_3$ with ultrasonication. After the cleaning
step, samples were transferred into a 1.5 ml tube, soaked in 0.5 ml Milli-Q water, and dissolved using



stepwise additions of 100 µl 0.5 M $HNO_3$ until the dissolution reaction was completed. The dissolved
samples were centrifuged, and the supernatant was immediately transferred to Teflon beakers to
prevent the leaching of any possible remaining phases. The solutions were then dried and Nd was
purified using Eichrom TRU-Spec and Ln-Spec resins following the analytical procedure described in
Copard et al., (2010). The $^{143}Nd/^{144}Nd$ ratios were measured using the Multi-Collector Inductively
Coupled Plasma Mass Spectrometer (MC-ICP-MS Neptune$^{Plus\ Thermo\ Fisher}$) (PANOPLY's analytical facilities
at the University Paris-Saclay, France) hosted at the *Laboratoire des Sciences du Climat et de*
*l'Environnement* (LSCE, Gif-sur-Yvette, France). Sample and standard concentrations were matched at
10 to 15 ppb, and mass fractionation was corrected by normalising $^{146}Nd/^{144}Nd$ ratios to 0.7219,
applying an exponential law. During the analysis, every group of three samples was bracketed with
analyses of JNdi-1 Nd standard solution, which is characterised by certified values of 0.512115 ±
0.000006 (Tanaka et al., 2000). The analytical error reported for each sample analysis is based on the
external reproducibility ($2\sigma$) of the JNdi-1 standard within a given session, unless the internal error
was higher. The Nd isotopic composition is expressed as $\varepsilon_{Nd}$ = [($^{143}Nd/^{144}Nd$)$_{Sample}$/($^{143}Nd/^{144}Nd$)$_{CHUR}$ - 1]
x 10,000, where ($^{143}Nd/^{144}Nd$)$_{CHUR}$ = 0.512638 represents the chondritic uniform reservoir (Jacobsen
and Wasserburg, 1980).

*3.6 Grain size analysis*
We measured grain size distributions using a Mastersizer 3000 at the University of Galway, Ireland,
School of Geography, Archaeology and Irish Studies. We applied a refractive index of 1.54 and an
absorption index of 0.01, as recommended by *Malvern Panalytical*. We used ~1 g of dry bulk
sediment; samples were pre-sieved at 1000 µm, and following Jonkers et al. (2015) we removed
carbonates, organic matter, and opal, before measuring. Challenges associated with this method
include air bubbles introduced to the optical cell within the Mastersizer during analysis, which can
skew the results towards larger grain sizes. To alleviate this problem, we cut grain size distributions at
211 µm. Since 99% of sediments are below this size bin (Polakowski et al., 2021) cutting grain size
distributions likely had no impact on sediments deposited by WTOW (i.e., silt ≤ 50 µm and clay ≤2 µm).
We then statistically unravelled grain size distributions into end-members using a non-parametric End-
Member Analysis (Weltje, 1997) in AnalySize (v. 1.1.2) (Paterson and Heslop, 2015). A non-parametric
approach estimates end-members from the dataset and does not rely on assumed knowledge of the
distribution, i.e., the size distributions of the subpopulations are not known a priori and must be
determined from the data itself (Paterson and Heslop, 2015;Chen and Guillaume, 2012). To ensure the
datapoints adequately represent the end-members, we excluded specimens with a $r^2$ <0.99. The





resulting end-members were assessed for their size, distribution and sorting. Well-sorted end-
members in the silt-sized fraction were interpreted as current-sorted sediments, which preserve a
bottom-water current speed signal (Prins et al., 2002), while poorly sorted end-members including
sand and gravel were interpreted as representing IRD.
*3.7 X-Ray Fluorescence analysis*
X-Ray Fluorescence (XRF) analysis was performed at the Integrated Ocean Drilling Program, Bremen
Core Repository, in Germany on DSDP Core 94-610A (0–38 mbsf) and 610B (24–34 mbsf). Data were
collected every 0.5 cm down-core with a slit size of 5 mm using a generator setting of 10 kV, 0.035 mA
and a sampling time of 10s directly at the split core surface using an XRF Core Scanner III (Avaatech)
that measures selected elements between Aluminium and Uranium. Calcium (Ca), and Titanium (Ti)
are common elements observed in marine sediments that can be used as palaeoenvironmental tracers
(Gebhardt et al., 2008;Van Rooij et al., 2007;Arz et al., 2001). Ca is primarily of biogenic origin (Solignac
et al., 2011), and reflects the presence of calcium carbonate ($CaCO_3$) tests of foraminifers and
coccolithophorids in the sediments (Rothwell, 2015). It is well-recognised that $CaCO_3$ records in the
Atlantic are related to Glacial-Interglacial cycles, with higher $CaCO_3$ concentrations during interglacials
(Balsam and McCoy Jr, 1987). Ca can also be sourced from detrital material but this is most relevant
in near-shore environments (Rebolledo et al., 2008) or in the IRD Belt (Ruddiman, 1977) during North
Atlantic Heinrich events (Hodell et al., 2008). Ti is primarily terrigenous sourced and forms the detrital
load (Haug et al., 2001). Here we use the log(Ti/Ca) as a proxy for evaluating relative variations in
lithogenic/biogenic content (Piva et al., 2008).

**4 Chronology**
The fundamental aim of age modelling is to construct meaningful time series with age-depth
relationships and report the associated errors (Breitenbach et al., 2012;Trachsel and Telford, 2017).
The robustness of any age model depends on the number of fixed dates and the associated
uncertainties (Telford et al., 2004). Astronomical tuning is a commonly used tool to build age models
on Pleistocene sediment records (Clemens, 1999). Typically, sections are tuned to the LR04 benthic
$\delta^{18}O$ stack record by Lisiecki and Raymo (2005) based on 65°N June insolation values. The age model
for site 610B was constructed using continuous 0.5 cm resolution XRF analysis performed on DSDP
sites 610A and 610B for the past 500 ka. The chronology of MIS 11 was further constrained using $\delta^{18}O$
values picked throughout the core. These $\delta^{18}O$ measurements were tuned to the benthic $\delta^{18}O$ record
of the well-dated Ocean Drilling Project (ODP) site 980 (McManus et al., 1999), and its LR04
chronology, which has an uncertainty of ±4 ka BP (Lisiecki and Raymo, 2005). For a detailed
description of tie points used to constraint site 610B over MIS 11, please consult Holmes et al. (2022)
and Figure 3. Here, we add a further tie point at 29.58 mbsf (Figure 3) by linking the mid-interglacial
maximum Np % from site 610B to the mid-interglacial peak in Np % from the well-dated site 983
(Barker et al., 2015).

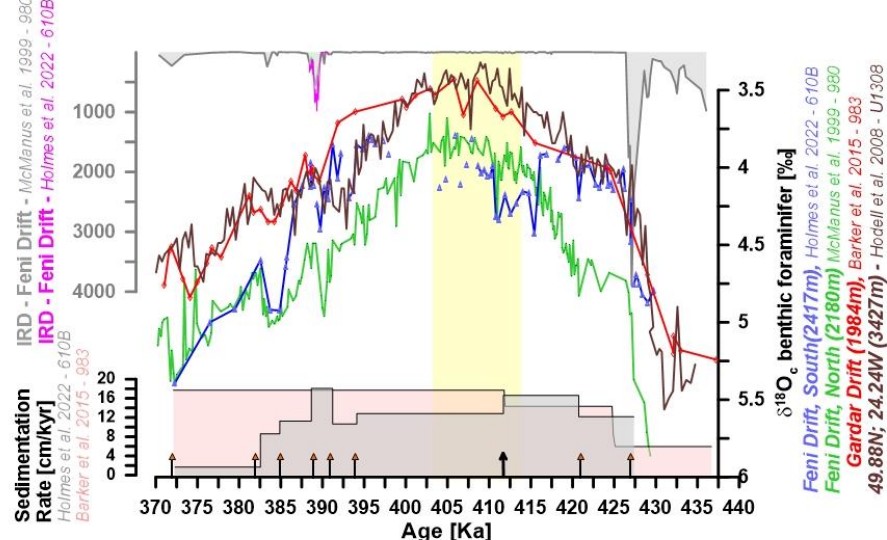

Figure 3. Age Model modified from Holmes et al. (2022). IRD counts from DSDP 610B (pink) and ODP 980 (grey)
from the Feni Drift; $\delta^{18}O$ benthic foraminifera records from DSDP 610B (blue), ODP980 (green, and ODP 983
(red), IODP 303-U1308 (brown) corrected by 0.63 ‰. Sedimentation rates for DSDP 610B (grey) in cm.kyr$^{-1}$ are
adjusted for the additional tie point used in this study (black arrow). Orange arrows show tie points used in
Holmes et al. 2002 and in this study. Also shown are sedimentation rates for ODP 983 (light pink). The yellow
band marks the dataset shown here.
We acknowledge that in doing so we assume that the associated cooling of the 412-ka event occurred
simultaneously across the subpolar North Atlantic. However, we believe that this approach is justified
since most records (surface and deep-water) from the North Atlantic Region place the main signal of
the ~412 ka event at 411.7 (±0.7) ka (Table 1). Further, we note that the duration of the surface cooling
observed in records near site 610B (e.g., sites 980, U1314, 983, M23414 (TEX$_{86}$)) is similar (0.4-0.9 ka)
regardless of the age model used. Considering the short duration of the 412-ka event, <1 ka, and the
uncertainties for chronologies based on $\delta^{18}O$ curves it would not be possible to assess the regional
progression (if there was one) for the event with or without tie points between chronologies. With
the inclusion of this additional tie point, the event occurred at 411.9 ka in core 610B, and the
presented record here covers the period from 414.3-403.4 ka and sedimentation rates within this
period correspond to ~82 years per cm.
Table 1: The event in surface and deep-water and the corresponding ages in the North Atlantic



### Surface water

| Site | Latitude | Longitude | Age (ka) | Depth (m) | Proxy | Reference |
|---|---|---|---|---|---|---|
| ODP 983 | 60°4 N | 23°6 W | 411.8 | 1984 | Np coiling ratio | Barker et al. (2015) |
| U1305 | 57°29 N | 48°32 W | 413.2 | 3459 | Pf % | Irvali et al. (2020) |
| U1314 | 56°21 N | 27°53 W | 412 | 2820 | Pf % | Alonso-Garcia et al. (2011) |
| ODP 980 | 55°29 N | 14° 42 N | ~411.1* | 2179 | Np % | Oppo et al. (1998) |
| M23414 | 53°32 N | 20°17 W | 411.3 | 2196 | TEX$_{86}$ | Kandiano et al. (2017) |

### Deep-water

| Site | Latitude | Longitude | Age (ka) | Depth (m) | Proxy | Reference |
|---|---|---|---|---|---|---|
| U1305 | 57°29 N | 48°32 W | 411.6 | 3459 | Ebf δ13C | Galaasen et al. (2020) |
| ODP 980 | 55°29 N | 14° 42 N | ~410.9* | 2179 | Ebf δ13C | McManus et al. (1999) |
| U1308 | 49°87 N | 24°23 W | 411.6 | 3427 | Ebf δ13C | Hodell et al. (2008) |

Pf – Planktonic foraminifera; Ebf – Epifaunal benthic foraminifera, *ODP Site 980 on the LR04 age model
To ensure an objective assessment of climate transitions for each data series (log(Ti/Ca),
log(EM2/EM3), and MAT derived SST) presented here, we applied a Ramp function using the Fortran
77 program, RAMPFIT (Mudelsee, 2000). This is a statistical programme that uses Brute-force to
estimate the unknown onset and end of a time interval by weighted least-squares regression to
determine the best fit. Following Tibshirani and Efron (1993) we use a bootstrap simulation of 200
resamples to estimate the uncertainty of the results (Table 2). To determine the ramps objectively the
search interval (x$_1$, x$_2$) was set as far apart as possible but before the next shift in climate state. This
enabled the programme to statistically determine the most significant ramp for the onset, duration,
and recovery for each dataseries. All search intervals are shown in Table 2.

Table 2: Rampfit for each proxy. Log(Ti/Ca) = lithogenic/biogenic variations, SST = Sea Surface Temperature,
WTOW = Wyville-Thomson Overflow Water. Rampfit is an autoregressive model used to describe time-varying
natural processes that accurately quantify transitions (Mudelsee, 2000). SE = standard error.

| Dataset | Interval | | Ramp 1 | SE | Ramp 2 | SE | Duration | SE |
|---|---|---|---|---|---|---|---|---|
| XRF Log(Ti/Ca) | 412.23 | 415.5 | 412.68 | 0.24 | 414.58 | 0.22 | 1.9 | 0.39 |
| | 410.96 | 412.68 | 411.99 | 0.02 | 412.29 | 0.01 | 0.3 | 0.03 |
| | 410.03 | 411.99 | 410.69 | 0.23 | 411.08 | 0.23 | 0.39 | 0.4 |
| | 410.69 | 409.6 | 409.83 | 0.09 | 410.03 | 0.08 | 0.2 | 0.15 |
| | 409.83 | 408.23 | 408.62 | 0.17 | 409.75 | 0.14 | 1.13 | 0.26 |
| | 408.62 | 404.18 | 404.3 | 0.15 | 407.45 | 0.15 | 3.16 | 0.23 |
| | 404.3 | 400 | 401.88 | 0.8 | 404.26 | 0.7 | 2.38 | 1.19 |
| WTOW | 410.3 | 414.28 | 411.9 | 0.28 | 412.86 | 0.45 | 0.96 | 0.66 |
| | 408.12 | 411.9 | 410.1 | 0.55 | 410.3 | 0.54 | 0.2 | 0.67 |
| | 403.4 | 410.1 | 406.01 | 1.62 | 410.1 | 1.39 | 4.09 | 1.76 |
| SST | 411.66 | 414.28 | 411.9 | 0.34 | 412.62 | 0.36 | 0.72 | 0.44 |
| | 409.91 | 411.9 | 411.27 | 0.24 | 411.9 | 0.1 | 0.62 | 0.26 |

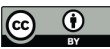

| $\delta^{13}C$ | 409.13 | 411.27 | 409.52 | 0.23 | 409.91 | 0.43 | 0.39 | 0.5 |
| --- | --- | --- | --- | --- | --- | --- | --- | --- |
| | 403.4 | 409.51 | 406.4 | 0.87 | 409.32 | 0.68 | 2.92 | 1.27 |
| | 413.85 | 408.90 | 412.11 | 0.71 | 412.17 | 0.56 | 0.06 | 0.83 |
| | 412.11 | 405.74 | 408.86 | 0.97 | 408.90 | 0.99 | 0.04 | 0.99 |
| | 4.08.90 | 403 | 405.47 | 0.50 | 405.74 | 0.74 | 0.27 | 0.93 |

## 5 Results

### 5.1 X-Ray Fluorescence (XRF) and Ice-Rafted Debris (IRD)

At 412.29 (±0.01) ka the contribution of lithogenic versus biogenic (e.g., log(ti/Ca)) input sharply increases over 0.3 (±0.03) ka (Figure 4). Terrigenous input remains elevated for another 0.39 (±0.40) ka before decreasing towards a mid-event plateau. A second maximum in terrigenous input occurs at 409.75 (±0.14) ka. Thereafter, values decrease to pre-event values by 404.30 (±0.15) ka. The overall structure of the time series is best described by a two-step event. IRD abundance is relatively low throughout the record, fluctuating between 2.7 and 119.6 grains/g. Nevertheless, the overall structure between the XRF and IRD

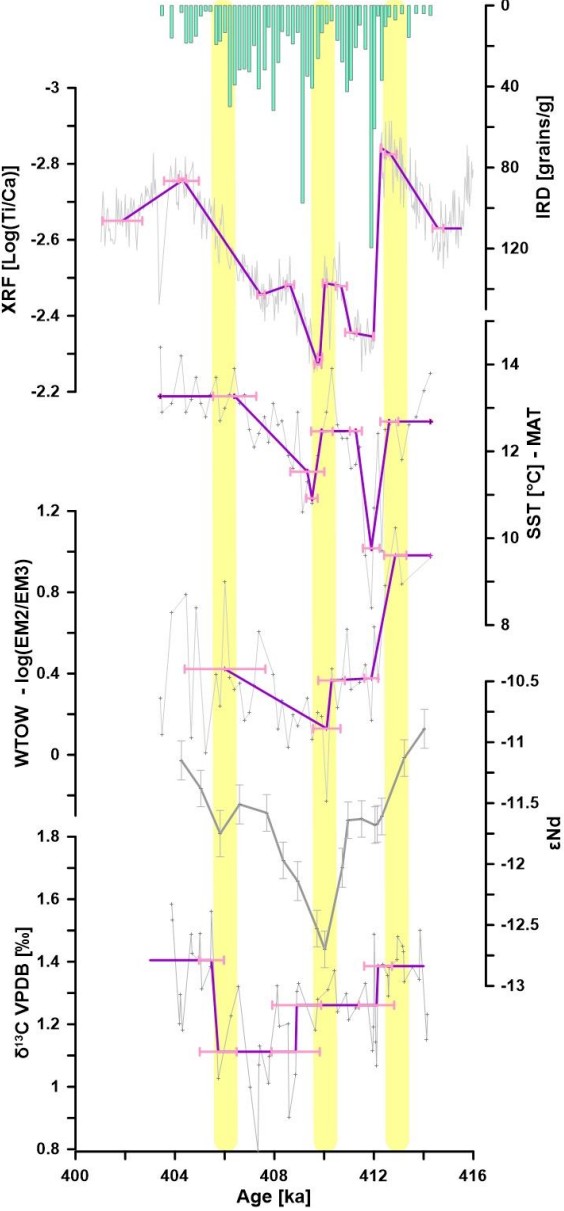

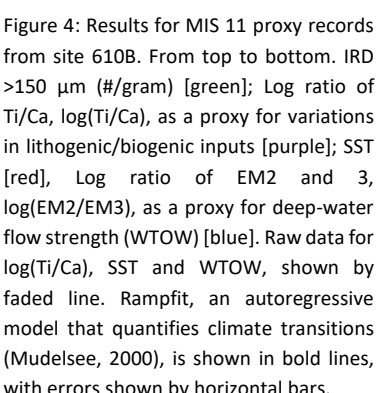
Figure 4: Results for MIS 11 proxy records from site 610B. From top to bottom. IRD >150 µm (#/gram) [green]; Log ratio of Ti/Ca, log(Ti/Ca), as a proxy for variations in lithogenic/biogenic inputs [purple]; SST [red], Log ratio of EM2 and 3, log(EM2/EM3), as a proxy for deep-water flow strength (WTOW) [blue]. Raw data for log(Ti/Ca), SST and WTOW, shown by faded line. Rampfit, an autoregressive model that quantifies climate transitions (Mudelsee, 2000), is shown in bold lines, with errors shown by horizontal bars.





records is similar with two distinct maxima centred at 411.90 and 409.52 ka, placing the maxima within
the periods of maximal lithogenic input, as
inferred from the XRF record (Figure 4).

*5.2 Sea surface temperatures (SST) and*
*foraminifera abundances (%)*
The two-step event structure is also
evident in the MAT-derived SST
reconstruction (Figure 4). Before the
event, between 414.3 and 412.62 (±0.36)
ka, SST values describe a period of
persistent warmth (e.g., 11.8-13.8 °C).
Over the same period, foraminifer
assemblages show relatively high
abundances (Figure 5) of transitional
species (i.e., *G. bulloides, G. glutinata*),
and an enhanced influence of the
subpolar species, *N. incompta*, reaching
maximum abundance, forming almost
50% of the total assemblage, just before
the first cooling event at 412.62 (±0.36)
ka. The first drop in SST of ~3.0°C from
12.7°C to 9.77°C occurred between 412.62
(±0.36) and 411.90 (±0.10) ka. A distinct
shift from subpolar/transitional species
during this period with minimum values
of *G. inflata* (4.5%), a sharp reduction in
*G. bulloides* (10.2%) and *N. incompta*
(39.8%) indicates a reduced influence of
warm, saline Atlantic Waters during this
time. SSTs recovered to 12.5°C before
decreasing a second time to 10.9°C at
409.51 (±0.23) ka. This second cooling is
concurrent with an increase of Np and an

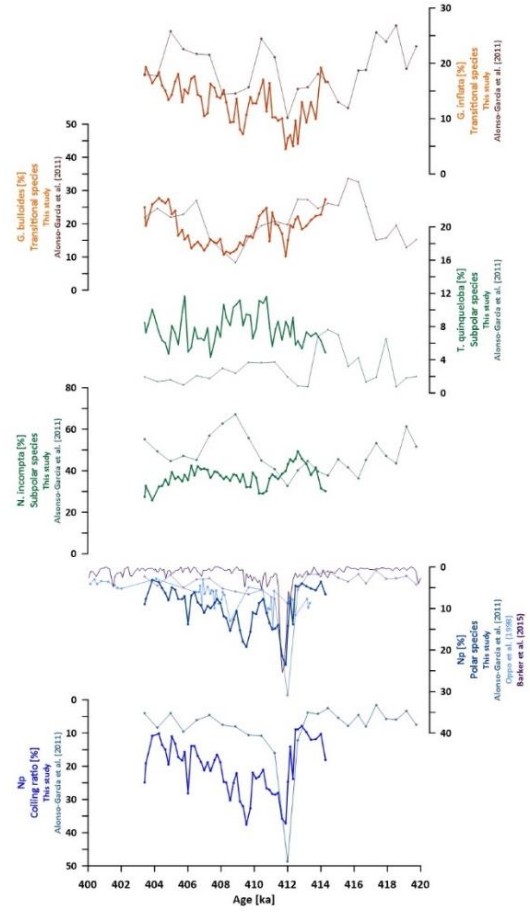

Figure 5 The evolution of planktonic foraminifera during peak MIS 11 conditions. From top to bottom, the relative abundances of G*. inflata* (transitional) [orange site 610B], [brown site U1314], *G. bulloides* (transitional), [orange site 610B], [brown site U1314], *T. quinqueloba* (subtropical), [dark green 459 610B], [light green site U1314] *N. incompta* (subtropical), [dark green site 610B], [light green site U1314], Np – note reversed axis (polar), [dark blue site 610B], [light blue site U1314], [baby blue, site ODP 980], [deep purple, site ODP 983] and Np coiling ratio (%) – note reversed axis, [dark blue site 610B], [light blue site U1314]. NOTE: OPD site 980 data have been updated to the LR04 age model. We exclude site U1305 data as the hydrographic setting is so different that the values do not easily plot with the data from the eastern North Atlantic. Please refer to the original publication for details (Irvalı et al., 2020).





increase in the Np coiling ratio to 37.6%. From then onwards SST slowly recovered, reaching higher
than pre-event values by 406.40 (±0.87) ka. Maximum values of 14.4°C occur towards the end of our
record at 403.40 ka. Both Np % and the Np coiling ratio track the two-step nature of the event as seen
in the XRF and SST records in terms of range of change and timing. The strong agreement between
IRD, SST, and XRF (Figure 4) supports the interpretation of the log(Ti/Ca) record as a climate indicator,
reflecting relative changes between IRD (i.e., lithogenic content) and climate (i.e., high $CaCO_3$
production during warmer climates).
*5.3 Grain size analysis*
Our grain size analysis indicates the sediments are
adequately described by three end-members, one
IRD (end-member 1; EM1), and two overflow (end-
members 2 and 3; EM2 and EM3), inferred from
both the $R^2$ and angular distance (θ) goodness-of-fit
statistics ($R^2 = 0.253$, θ = 2.3°) (see also Supplement
Figure S1). The well-sorted EM2 and EM3 consist
primarily of clay and fine silt sediment with mean
grain sizes of 5.21 µm and 9.86 µm, respectively,
characteristic of sediments sorted by bottom-water
currents (Figure 6). The log ratio of EM2 and EM3
(log(EM2/EM3)) is used as a proxy for WTOW flow
strength in the Rockall Trough (Prins et al., 2002).
This ratio describes the relative increase in grain
size of EM2 over EM3 and thereby a decrease in
values infers a decrease in flow strength (Figure 4
and 6). EM1 is poorly sorted and is composed of
49.4 % clay, 50.3 % silt, and 0.32 % sand (Figure 6).
The relatively high proportion of clay and silt in EM1
suggests that EM1 most likely represents IRD
(Andrews, 2000;Jonkers et al., 2012;Nürnberg et
al., 1994). The highest contribution of the two, well-
sorted end-members (log(EM2/EM3)) occurs in the
oldest part of our record between 414.28 and
412.86 (±0.45) ka when IRD is low. At 412.86 (±0.45)

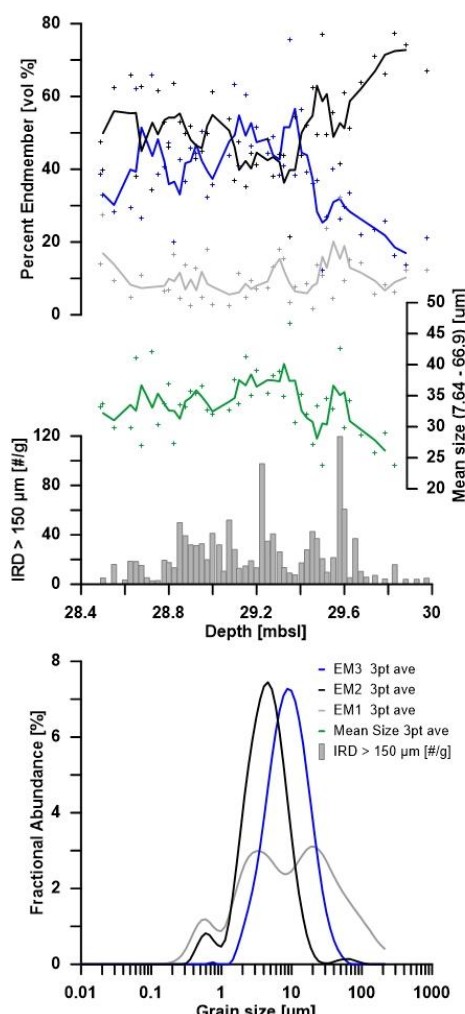

Figure 6 Grain size analysis. From the bottom up: The bottom graph shows the grainsize distribution of the main endmembers EM1(grey), EM2(black), and EM3(blue). This is followed by IRD >150 (grey bars), the mean grainsizes (7.64-66.9 µm) for each sample, and the proportion of each endmember (EM1-grey, EM2-black, and EM3-blue) plotted versus depth.

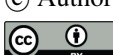



ka WTOW begin to decrease over 0.96 (±0.66) ka before stabilising between 411.90 (±0.28) and 410.30
(±0.54) ka. At 410.30 (±0.54) ka a second decrease in values occurs over 0.20 (±0.67) ka. Thereafter,
overflows slowly recover but remain below pre-event values and are variable until the end of the
record.
*5.4. εNd and stable isotopes ($\delta^{18}$O and $\delta^{13}$C)*
At 414 ka $\epsilon_{Nd}$ values are most radiogenic at -10.9 ± 0.2. This is followed by a decrease to less but stable
εNd values of -11.6 ± 0.2 between 412.31 and 410.92 ka. After 410.92 ka, $\epsilon_{Nd}$ values decreased a
second time to reach the lowest value of -12.7 ± 0.15 at 410.02 ka. Thereafter $\epsilon_{Nd}$ values increased
steadily until 407.68 and then again after 405.04 ka to reach -11.15 ± 0.15 at the end of the dataset
(Figure 4). We note that due to the limited sample set, we were not able to perform rampfit functions
to support the interpretation of the onset and recovery of the event with statistics in the $\epsilon_{Nd}$ dataset.
Both $\delta^{18}$O and $\delta^{13}$C values were high at 3.9-4.0 ‰ and 1.3-1.5 ‰ respectively at 414 ka. Following a
two-step pattern $\delta^{13}$C decreased first to 1.26 ‰ between 412.16 ± 0.56 and 408.89 ± 0.99 ka followed
by a second decrease to 1.12 ‰ between 408.86 ± 0.97 and 405.74 ± 0.74 ka. We note that variability
is high in this interval and notably $\delta^{13}$C values decrease to reach low values of 0.79 ‰ at 407.34 ka.
The recovery to more enriched $\delta^{13}$C values began after 405.47 ± 0.50 ka to reach 1.41‰ which are
higher than pre-event values.  $\delta^{18}$O values steadily decreased by 0.5-0.7 ‰ from 3.9-4.0 to 3.3-3.4 ‰
over the 10-ka analysed here.
**6  Discussion**
*6.1. The 412-ka event*
Sea surface temperature records across the Northeast Atlantic, from the Gardar Drift to the Rockall
Trough (e.g., sites 983, U1314, M23414, 980 and this study), record warm Holocene-like sea surface
conditions before ~412 ka (Kandiano et al., 2012). However, between 412.62 (±0.36) and 411.90
(±0.10) ka sea surface data at site 610B shows an increase in Np abundance (4.8 to 23.5%) and a drop
in SST of 3.0°C from 12.7°C to 9.77°C (Figure 4 and 5). Similarly, at sites 983, 980, and U1314 polar
species increase at ~412 ka (Figure 5). At site M23414, both mid-depth (TEX$_{86}$) and sea surface records
(Alkenone) also record a decrease in SST of 5.7°C in TEX$_{86}$ and 1.5°C in Alkenone SST at 412 ka (Figure
7) (Kandiano et al., 2017a). However, the event is absent in planktonic assemblage-based SST records
at the same site (Kandiano and Bauch, 2007) which is puzzling given the strong regional signal for the
event. A review of the methods used for SST reconstructions in Kandiano and Bauch (2007) reveals
that the foraminifera - based SST dataset was derived by combining and averaging three different
methods to infer summer SSTs using: Transfer Function Technique (TFT; Imbrie and Kipp (1971), MAT;



Prell (1985)) and Revised Analogue Method
(RAM; Waelbroeck et al. (1998)). Using the
average may have smoothed out the event in the
resultant SST data series.
Further west on the Eirik Drift warm conditions
also prevailed just downstream of the East
Greenland Current (EGC) (e.g., site U1305, Figure
2) with Np coiling ratios of 41.9-83.7% and SST
near 10°C from 420 ka until ~413.5 ka (Figure 7)
(Irvalı et al., 2020). Unlike reconstructions from
the Northeast Atlantic, these data provide
evidence for a much warmer sea surface climate
when compared to early Holocene and modern
core top (e.g., 7.7°C) values downstream of the
EGC (Irvalı et al., 2020). This long period of
warmth is interrupted by a two-step cooling
event of 7.2°C, from 10°C down to 2.8°C between
413.2-411.4 ka (Irvalı et al., 2020) (Figure 7). The
cooling is coeval with Np abundance and coiling
ratio reaching 96.5% and 100% respectively, and
occurred over approximately 0.8 ka (Irvalı et al.,
2020). On the Gardar Drift (site U1314) the Np %
increase is also of high-magnitude (from 3.8% to
48.6%) and occurs over 2 samples representing
1.22 ka (Alonso-Garcia et al., 2011). We note that
the duration of these transitions is limited by the
resolution of the respective archives and are thus
maximum estimates.

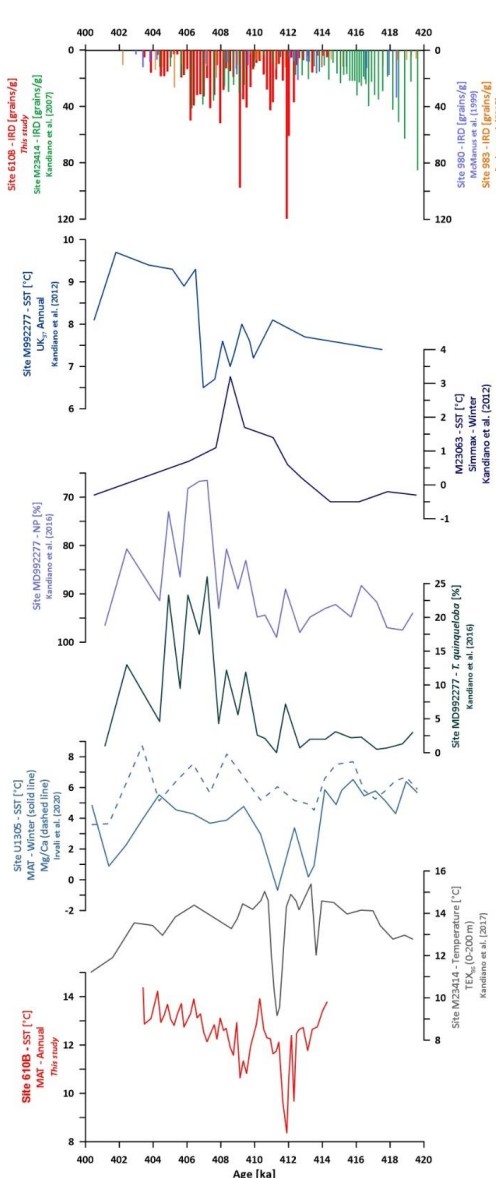

Figure 7: Surface hydrographic conditions during MIS 11. IRD grains/g records sites: site 610B [red] this study; site 980 [blue] (McManus et al., 1999); site M23414 [green] (Kandiano and Bauch, 2007) on the updated age model from Kandiano et al. (2017b) and site 983 [orange] (Barker et al., 2015). SST [°C]/ foraminifera assemblages [%] from top to bottom: Annual SST (UK37), site MD992277 (Kandiano et al., 2012); Winter SST (Simmax), site M23063 (Kandiano et al., 2012); % *N. pachyderma* – Np – note reversed axis, site MD992277 (Kandiano et al., 2016); *T. quinqueloba*, site MD992277 (Kandiano et al., 2016); SST (Mg/Ca), site U1305 [dashed line], Winter SST (MAT) site U1305 [solid line], (Irvalı et al., 2020); SST TEX86 (0-200 m) site M23414 (Kandiano et al., 2017b); MAT SST, site 610B (this study).



In the Nordic Seas, the 412-ka event trajectory seems to have been reversed (Figure 7). During early
MIS 11 (e.g., 424-412 ka) evidence for cold and fresh surface waters in the Iceland and western
Norwegian Sea is widespread. For example, at site M23352 in the North Iceland Basin (Figure 2),
continuous IRD during early MIS 11 (Helmke and Bauch, 2003), indicate that cold waters dominated
this core site until ~407.5 ka. Similarly, at site M992277 (sometimes referred to as PS1243) located in
the central Nordic Seas (Figure 2), high percentages of Np (81-99%) imply cold conditions until ~407
ka (Kandiano et al., 2012). Further east, at site M23063, cold temperatures prevailed until ~412.8 ka
(Kandiano et al., 2012). A comparison of foraminifera abundance-derived temperatures between the
first half of MIS 11 and the early Holocene (11-5.5 ka) at site M23063 illustrates the sharp difference
between the much warmer Holocene (4.12°C) and cold early MIS 11 (0.1°C), emphasising the unique
hydrographic conditions in the central Nordic Seas at the time (Kandiano et al., 2012). These datasets
also illustrate that cold waters reached much further east in the central Nordic Seas than during the
early Holocene (Doherty and Thibodeau, 2018; Kandiano et al., 2017b). Multiple studies hypothesise
that continuous background melting of the GIS especially south of 69°N contributed to these cold sea
surface conditions until 412 ka (de Vernal and Hillaire-Marcel, 2008; Reyes et al., 2014b; Robinson et
al., 2017b; Robinson et al., 2012; Willerslev et al., 2007). Mid-way through MIS 11, SSTs transition to
warmer more Holocene-like conditions. The exact timing for the transition from cold to warmer SSTs
is difficult to determine given the low resolution of records from the Nordic Seas and chronological
constraints, however, the general timing and trends in surface records appear to coincide with the
cold event in the subpolar North Atlantic at ca 412 ka.

What was the cause for the sudden sea surface cooling across the subpolar gyre at 412 ka? The faunal
assemblage changes and inferred SST cooling of 7°C at site U1305, describe a rapid transition from
Atlantic to Polar Waters immediately downstream of the EGC (Figure 5). Specifically, the decrease of
*T. quinqeloba* while Np % is increasing is indicative of the passage of the SAF and Polar Front over the
site (Alonso-Garcia et al., 2011;Mokeddem et al., 2014). This rapid transition in foraminifera
assemblages is also recorded at sites U1314 and 983 albeit at lower magnitudes invoking an eastward
progression of cold and potentially freshwater, together with the SAF, across subpolar latitudes during
the event.

A potential source of ice or cold/freshwater could have come from residual continental Ice during
early MIS 11. Indeed, most estimates of sea level rise following Termination V (Rohling et al., 2010;
Elderfield et al., 2012; Grant et al., 2014; Shakun et al., 2015; Spratt and Lisiecki, 2016; Giaccio et al.,
2021), agree that about 50–80 m SLE remained present in ice sheets near the start of MIS 11c due to



weak caloric summer insolation at 65˚N at 424 ka. By 412 ka estimates still assume the presence of
between -38.7 to +3.9m SLE of continental Ice (Sprat and Lisiecki et al. 2016, Grant et al. 2014, Shakun
et al. 2015, Elderfield et al. 2012). These estimates invoke the presence of several Greenland
equivalent size Ice sheets present at 412 ka. However, there is little geologic or terrestrial evidence to
support the presence of continental ice of this scale at high northern latitudes by 412 ka. For example
terrestrial palaeoclimate records from Europe (Tzedakis et al., 1997;Reille et al., 2000;Tzedakis et al.,
2006;Ashton et al., 2008;Preece et al., 2007;Nitychoruk et al., 2005) and Siberia (Prokopenko et al.,
2010;Melles et al., 2012;SHICHI et al., 2009) provide evidence for increased forestation at high
northern latitudes replacing tundra or frozen soils in Eurasia, driven by a lengthening of the growing
season during the obliquity maximum at 416 ka. Model simulations further suggest that the decrease
in summer sea ice at this time led to warming feedback during the winter at high Northern latitudes
amplifying the insolation-induced warming (Kleinen et al. 2014). Similarly, pollen-based
reconstructions (Melles et al., 2012;Prokopenko et al., 2010;SHICHI et al., 2009;Desprat et al.,
2005b;Nitychoruk et al., 2005) support warmer than present summers and winters in the High Arctic
Region throughout MIS11 until at least 405 ka.

A recent investigating into the retreat of the Laurentide Ice Sheet (LIS) following the glacial
Termination of MIS 12 (e.g., T5) also shows that the LIS would have been mostly deglaciated with the
Hudson Bay Ice Saddle collapse at 419 ± 4.7 ka (Parker et al., 2023). There is a possibility that ice
remained on land (e.g., Keewantin Ice Dome and/or Quebec-Labrador Ice Dome) until ca 405 ± 4.7 ka
(Parker et al., 2023) contributing to background melting via Hudson Bay from 419-405 ka. While
background melting is possible, the relatively low IRD input throughout the North Atlantic region
following the Hudson Bay Ice Saddle collapse and during the 412-ka event make it unlikely that a
collapse of marine-terminating ice shelves or glaciers from the LIS was involved in cooling the subpolar
North Atlantic. Indeed, IRD of 2.7-119.6 grains.g$^{-1}$ at site 610B throughout the event are comparable
to Holocene values of 4.7-113.9 grains.g$^{-1}$ recorded from the Feni drift at site 980 (McManus et al.,
1999). Similarly, IRD counts from Gardar Drift at site 983 (Barker et al., 2015) and from Eirik Drift at
site U1305 (Irvalı et al., 2020) are also low and comparable to Holocene values. Thus, delivery of
Icebergs remained low during the 412-ka event suggesting a distal source for calving icebergs at best.
Finally, if background melting via Hudson Bay persisted over the 419-405 ka period, we note that it
does not seem to have significantly impacted subpolar North Atlantic SSTs given the widespread
Holocene-like SSTs across mid-latitudes before the 412-ka event.



Another possibility for the presence of continental Ice may be an ice sheet northwest of Greenland
restricting the Canadian Arctic Archipelago until 412 ka. If so, this could have channelled all freshwater
exports via Fram Strait into the Nordic Seas (Lofverstrom et al., 2022) and thereby contributed to the
anomalously cold SST observed from the Nordic Seas until 412 ka. Once open, Arctic freshwater would
have been channelled south via both Baffin Bay and Fram Strait reducing freshwater export into the
Nordic Seas. We note that this scenario could explain the warming trend observed in the Nordic Seas
around the 412-ka mark. However, the presence of land ice in the Canadian Arctic Archipelago until
412 ka is difficult to reconcile with the evidence for high latitude warming from terrestrial and
modelling evidence and the full deglaciation of Camp Century (77.17°N 61.13°W) by 416 ± 38 ka.
(Christ et al., 2023). Furthermore, while dating uncertainties are large, there is evidence that Camp
Century remained fully deglaciated for at least 16 ka, which must precede the glacial inception at 397
ka and therefore 412 ka. Finally, models and terrestrial records indicate a completely ice-free southern
and western Greenland, except for highly elevated areas, by 411 ka (Robinson et al., 2017b;Robinson
et al., 2012). Low RSL estimates for early MIS 11 based on benthic foraminifera $\delta^{18}O$ records thus seem
to stand alone against multiple lines of evidence suggesting that high northern latitudes were
experiencing warmer climates and less ice. Nevertheless, we cannot rule out that land ice persisted
and contributed to the event given the chronological uncertainties associated with palaeo records.

*6.2 Deep-water response to the 412-ka event.*
The leading hypothesis describing a strong AMOC during early MIS 11 posits that a strong density
gradient between the subpolar North Atlantic and the Nordic Seas supported strong deep-water
formation in the Nordic Seas (Kandiano et al., 2012;Doherty et al., 2021). Our data supports the
presence of strong Nordic Sea deepwater production before 412 ka. First, grain-size inferred current
velocities preceding the event are among the highest recorded throughout MIS 11 including the glacial
inception (Holmes et al., 2022). Further, both $\delta^{18}O$ and $\delta^{13}C$ values are enriched at 3.9-4.0 ‰ and 1.3-
1.5 ‰ respectively, indicating a strong nutrient-poor northern source of deep waters such as
ISOW/WTOW from the Nordic Seas before the event (Figure 4). Similarly, the radiogenic $\epsilon_{Nd}$ signal of
-10.89 supports a strong influence of ISOW/WTOW (−10.3 ± 0.2) (Dubois-Dauphin et al., 2017) at the
site (Figure 4). In combination with high WTOW flow endmembers, this supports vigorous export of
NSDW via the Wyville-Thompson Ridge into the Rockall Trough before the 412-ka event.
Starting at 412.86 (±0.45) ka grain-size inferred current velocities describe a two-step decrease in flow
strength. Considering the temporal uncertainties, this reduction in overflow is concurrent with the SST
decreases at 412.62 (±0.36) ka. However, the higher resolution log(Ti/Ca) data records the change in
surface ocean properties at a multi-centennial delay with respect to the overflows at 412.29 (±0.01)
ka. This delay occurs within uncertainties of the SST record but is significant compared to the
overflows. This would suggest that there is a significant offset between the surface and deep-water
response to the cooling event recorded at site 610B. In effect, this highlights that deep water
circulation responds to cooling at subpolar latitudes before the eastward progression of cold water
reaches the eastern North Atlantic.
Concurrent with the two-step decrease in WTOW flow estimates, $\varepsilon_{Nd}$ values become less radiogenic
reaching the lowest values of -12.7 ± 0.15 at 410.02 ka (Figure 4). Together, these data suggest a
decrease in flow strength and a decrease in ISOW/WTOW contribution, replaced by either the less
radiogenic eNd Lower NADW (−12.1 ± 0.2 and 13.1 ± 0.2) or LSW (−13.4 ± 0.3 and −14.0 ± 0.3) (Dubois-
Dauphin et al., 2017;Lambelet et al., 2016) (Lambelet et al., 2016;Dubois-Dauphin et al., 2017) at site
610B. A larger contribution of LNADW or LSW is more likely than a larger contribution of
Mediterranean Overflow Water (MOW) or Southern sourced Ocean Waters (SOW) at this time
because both have an $\varepsilon_{Nd}$ signature of -11, which is too radiogenic to explain the excursion observed
in the data.
Following the lowest flow rates and reduced WTOW contribution at 410.02 ka benthic foraminiferal
$\delta^{18}$O and $\delta^{13}$C values at 610B at site 610B decrease (see also Supplement Figure S2), suggesting a
possible increased influence of SOW, LDW or/and LSW at site 610B than before the event. This is also
mirrored in the paired benthic $\delta^{18}$O and $\delta^{13}$C values from the Eirik Drift at site U1305 (Galaasen et al.,
2020) and at site U1308 (Hodell et al., 2008) located to the southwest of the Rockall Trough. At Eirik
drift the event is marked by a drop in benthic $\delta^{13}$C values of ca. 0.6 ‰ from 0.80 to 0.20 ‰. This
decrease, while smaller, is reminiscent of the 1 ‰ decrease observed during the 8.2 ka event when
glacial Lake Agassiz drained into the subpolar North Atlantic during the early Holocene (Kleiven et al.
2008). Kleiven et al. 2008 suggested that this decrease in benthic $\delta^{13}$C values describes the
replacement of low-nutrient LNADW supplied by DSOW with a high-nutrient deep-water mass from a
southern source (e.g., SOW) at the core site.
A shoaling and northwards extension of SOW into the Rockall Trough may also be a plausible response
to the decrease in NSDW formation at 412 ka. Modern SOW is characterised by $\varepsilon_{Nd}$ values near -11
(Dubois-Dauphin et al., 2017) and depleted $\delta^{13}$C values of 0.0 - 0.2 ‰ (Eide et al., 2017). The return to
more radiogenic $\varepsilon_{Nd}$ values shortly after 410 ka while both WTOW flow speeds and $\delta^{13}$C remain
depleted/variable could therefore be linked to a larger contribution of SOW at site 610B rather than
a rapid return of WTOW. A more northern influence of SOW during the event is also supported by a
0.6 ‰ drop in $\delta^{13}$C from ca 1.1 to 0.5 ‰ at site U1308 (south of site 610B) (Figure 8). We note that no
significant change in $\delta^{13}$C is observed at the more northern site 980 over this timeframe, suggesting





that the northward extend of SOW was perhaps limited to the southern Rockall Trough. Alternatively,
the deeper location of site 610B relative to site 980 (ca. 300m) may describe the depth boundary of
SOW influence during the event.
By 405.74 ka benthic carbon isotopes return to more enriched values. This trend is also seen at sites
U1308 and U1305 (Galaasen et al., 2020, Hodell et al. 2008) (Figure 8). $\varepsilon_{Nd}$ values return to ca. -11.15
altogether suggesting a return of WTOW at site 610B, however, the highly variable nature of flow
speeds in our WTOW flow record suggests that southward flow might have been intermittent at this
time – perhaps similar to modern observations (Johnson et al., 2017). The slow recovery of WTOW
current velocities seems to coincide with the slow retreat of cold waters towards the SPG. For
example, at site 610B SSTs only reach pre-event values by 406.40 (±0.87) ka and warmest SST by the
end of the dataset at 403.40 ka.
Similarly, Np % decreases slowly across
the SPG (e.g., sites U1314, U1305, 980)
to reach pre-event values by ~406 ka
(Alonso-Garcia et al., 2011;Irvalı et al.,
2020;Oppo et al., 1998). We also note
that the variability and continued
presence of some minor IRD at site
610B coincides with lower benthic $\delta^{13}$C
throughout the record, suggesting a
relationship between continued
iceberg rafting and deep ventilation
until 405.74 ka. The faster recovery
observed in $\delta^{13}$C values shortly after
the event at sites 980 and M23414,
may again be linked to the shallower
depth of both sites relative to site
610B, and delimit the varying depth
boundaries of water masses during the
perturbation.

6.3. Climate forcing and Ocean-
Atmosphere teleconnections.

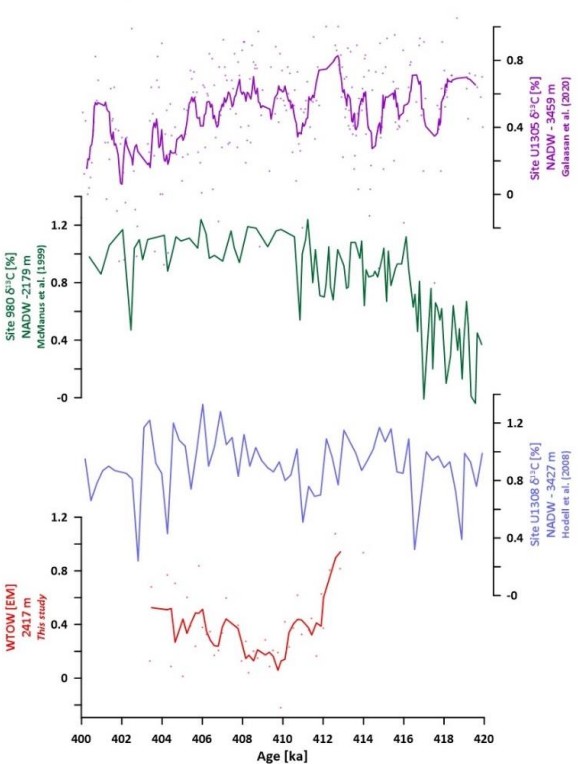

Figure 8: Deep-water records from North Atlantic Region. Top to bottom, $\delta^{13}$C – water mass proxy – site U1305 [purple] (Galaasen et al., 2020); site 980 [green] (McManus et al., 1999); site U1308 [light blue] (Hodell et al., 2008);, WTOW grain size analysis– deep-water strength proxy site 610B [red] (this study). NOTE: OPD site 980 data updated to the LR04 age model.





A strong AMOC seems at odds with the western Nordic Seas covered by meltwater, especially since
strong deep-water formation in the Nordic Seas requires Atlantic inflow and open-ocean convection
(Eldevik et al., 2014). However, evidence for a strong AMOC particularly coinciding with the peak in
obliquity (Kessler et al., 2020) at 416 ka is supported by several model simulations (Rachmayani et al.,
2017) and palaeoceanographic reconstructions from the North (Galaasen et al., 2020) and South
Atlantic (Dickson et al., 2009;Dickson et al., 2010). Revisiting the location and datasets used to infer
the cooling in the Nordic Seas may allow us to reconcile model simulations and observations. The
three sites from the Nordic Seas covering MIS 11 are located in the Iceland Sea and the western
Norwegian basin between 65-70°N. Today these sites are mostly influenced by the Jan Mayen Current
and the East Icelandic Current recirculating Arctic Waters from the EGC in the Iceland Sea Gyre. We
also note that the datasets inferred from these sites are based on foraminifera and Alkenones, both
known to represent a seasonally constrained summer signal especially at these high northern latitudes
(Fraile et al., 2009;Kretschmer et al., 2018;Bendle et al., 2005).

A series of time slice simulations across MIS-11 suggest that boreal summers were particularly warm
around Western Greenland, notably over the Canadian Arctic Archipelago and the high latitudes of
the Atlantic sector for a period of at least 10 ka from 418-408 ka leading to weakened high latitude
winds and the emergence of a single, unified midlatitude jet stream across the North Atlantic sector
during boreal summers (Crow et al., 2022). Similarly, Rachmayani et al. (2017) simulate a negative
southerly wind anomaly along the east Greenland margin centred over the Denmark Strait for MIS 11
in comparison to MIS 5e. If correct this more zonal and weaker circulation pattern might have led to
reduced export of cold waters out of the Nordic Seas on a seasonal basis. Reduced meridional wind
forcing during MIS11 summers (e.g., Crow et al. 2022) and an enhanced seasonal recirculation of
meltwaters in the Iceland gyre (Le Bras et al., 2018) may therefore be plausible mechanisms explaining
the "cool" Nordic Seas and "warm" subpolar North Atlantic signals.

Interestingly, the negative southerly wind anomaly centred over the Denmark Strait for MIS 11 is also
associated with an increased advection of salt from the south to the eastern North Atlantic
(Rachmayani et al., 2017) supporting strong NADW formation. Enhanced northern salt advection and
strong NADW formation during times of enhanced GIS melting have also been simulated in response
to RCP8.5 emission scenarios (Berk et al., 2021). Where the freshwater-induced weakening of
Labrador Seawater formation is compensated by a strengthening of NSDW formation similar to
observations made in (Wood et al., 1999;Swingedouw et al., 2013). Mechanistically, it is the weakened
subpolar gyre that leads to a shift of the North Atlantic Current and subpolar-subtropical gyre



boundary, with the subtropical gyre expanding and the subpolar gyre contracting (Swingedouw et al.,
2013;Berk et al., 2021). Under this scenario, it is possible that Atlantic Waters reached the northern
Nordic Seas pre-412 ka, perhaps in the form of a narrow boundary current along the Norwegian
continental margin contributing to high latitude warming and deepwater formation while the Icelandic
Sea remained cool as also observed during the early Holocene (Risebrobakken et al., 2011;Telesiński
et al., 2022).

We cannot rule out that the collapse or melting of remnant continental ice caused the 412-ka event,
however, the low IRD counts associated with the event do not support Ice rafting. Instead, we
hypothesise that a change in the seasonal balance of freshwater export from the Nordic Seas into the
North Atlantic via the Denmark Strait might have initiated a reorganisation of the freshwater
distribution at polar/subpolar latitudes. This could have led to a shift of the subpolar-subtropical gyre
boundary and thereby the northward advection of Atlantic Waters and cooling across the North
Atlantic Region. The difference between the recirculation of polar waters and "freshwater hosing"
scenarios, in which the thermohaline circulation weakens or collapses (Stouffer et al., 2006;Vellinga
et al., 2008;Drijfhout, 2015), therefore appears to be linked to freshwater reaching the Nordic Seas
via the Eastern North Atlantic (e.g., NAC). This has been hypothesised previously for future climate
simulations (Berk et al., 2021) and demonstrated for glacial boundary conditions (Muschitiello et al.,
2019) and the glacial inception following MIS11 (Holmes et al., 2022).  We now show that similar
freshwater (or sensitivity of thermohaline circulation) dynamics may have been in operation for full
interglacial boundary conditions similar to our pre-industrial climate.

The evidence presented here highlights the sensitivity of  NSDW formation to the reorganisation of
freshwater across the polar/subpolar boundary and the movement of the subpolar-subtropical gyre
boundary and therefore supports the recent hypothesis that high-magnitude variability of the AMOC
may not require major additions or outbursts of freshwater (Galaasen et al., 2020). Instead, the
reorganisation of Atlantic vs Polar waters at subpolar latitudes (Sgubin et al., 2017b) and thereby the
density gradient across the SPG (Müller et al., 2015) and between the subpolar North Atlantic and the
Nordic Seas (Jungclaus et al., 2006b) may determine high magnitude interglacial variability and
strength in overturning (Olsen et al., 2008;Hansen and Østerhus, 2000;Østerhus et al.,
2001;Mauritzen, 1996). For example, the variability in WTOW current velocities after the 412-ka event
illustrates a higher sensitivity of the overflows to the weaker density gradient between the Nordic
Seas and the subpolar North Atlantic modulated by a stronger east-west oriented SPG pushing the
subpolar-subtropical gyre boundary and with-it icebergs/freshwater into the eastern north Atlantic.





We note that state jumps (or hysteresis) in gyre circulation and exchange have also been found in
simple (Born and Stocker, 2014) and coupled models (Born et al., 2013) and can be associated with a
collapse in deep convection and reorganisation in AMOC geometry (e.g. where deep water is formed)
(Sgubin et al. 2017).

**7 Conclusion**
Prolonged interglacial warmth during MIS 11 especially at high northern latitudes led to intensive
background melting and the demise of continental ice sheets, including the southern and western
Greenland Ice Sheets (GIS) by ~412 ka. Despite the addition of freshwater to the Nordic Seas, Deep-
Water formation there remained strong, sustained by a strong density gradient between the Nordic
Seas and the subpolar North Atlantic.
The abrupt reorganisation of freshwater into the Subpolar Gyre (SPG), and expansion into the eastern
North Atlantic at 412-ka, reduced the density gradient between the Nordic Sea and the SPG and
thereby the inflow of Atlantic Waters into the Nordic Seas. In addition, continuous freshwater export
into the SPG may have further subdued deep-water flow in the western North Atlantic, for the
remainder of the interglacial.
The evolution of surface and deep-water circulation described here sheds important insights on the
sensitivity of deep-water formation in response to continuous background melting from the GIS during
similar or warmer than present climate boundary conditions. Our examination reveals that the
reorganisation between Polar and Atlantic Waters at subpolar latitudes is central mechanistically, not
only for intermediate (Muschitiello et al., 2019), and low-ice (Holmes et al., 2022) boundary conditions
but also for warm interglacial climates such as MIS 11 and the current Holocene.
Our findings demonstrate that the availability and/or rate (Lohmann and Ditlevsen, 2021) of
freshwater reaching subpolar latitudes is modulated by non-linear atmosphere-ocean feedbacks (e.g.,
rate-induced tipping point), regardless of boundary conditions. This is crucial given current and
projected GIS melting, and the estimated sensitivity of the AMOC to surface water buoyancy
fluctuations (Smeed et al., 2014;Thornalley et al., 2018;Caesar et al., 2018;Bakker et al., 2016;Yu et al.,
2016). This study concludes that continuous freshwater input alone is unlikely to inhibit NSDW
formation and that a reorganisation of surface waters at subpolar latitudes is fundamental to overflow
strength.

**Data and materials availability**



All data needed to evaluate the conclusions in the paper are presented in the paper and/or the
Supplementary Materials. Raw data will be made available in Pangaea upon publication.

**Author contribution**
The research and iCRAG-GSI Environmental Geoscience proposal was designed and
managed by AM in collaboration with UN and CC. MC performed faunal counts, sediment
size analysis, IRD counts, XRF scans, and data analysis and wrote the first draft of the
manuscript. UN performed stable isotope analysis. CC and MMOC performed Nd analysis.
AM wrote the final version of the manuscript with contributions from UN and CC.

**Competing interests**
The contact author has declared that neither they nor their co-authors have any competing
interests.

**Acknowledgements**
We gratefully acknowledge the assistance provided by Arnaud Dapoigny and Louise Bordier
during Nd isotope analyses.

**Financial support**
This research has been supported by the iCRAG-GSI Environmental Geoscience PhD
Programme (17/RC-PhD/3481) awarded to AM and MMOC and the Galway Fellowship
awarded to MC.

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

Distribution of Various Soil Materials Measured by Laser Diffraction—The Problem of
Reproducibility, Minerals, 11, 465, 2021.
Praetorius, S. K., McManus, J. F., Oppo, D. W., and Curry, W. B.: Episodic reductions in bottom-water
currents since the last ice age, Nature Geoscience, 1, 449-452, 2008.



Preece, R., Parfitt, S., Bridgland, D., Lewis, S., Rowe, P., Atkinson, T., Candy, I., Debenham, N.,
Penkman, K., and Rhodes, E.: Terrestrial environments during MIS 11: evidence from the Palaeolithic
site at West Stow, Suffolk, UK, Quaternary Science Reviews, 26, 1236-1300, 2007.
Prell, W. L.: Stability of low-latitude sea-surface temperatures: an evaluation of the CLIMAP
reconstruction with emphasis on the positive SST anomalies. Final report, Brown Univ., Providence,
RI (USA). Dept. of Geological Sciences, 1985.
Prins, M. A., Bouwer, L. M., Beets, C. J., Troelstra, S. R., Weltje, G. J., Kruk, R. W., Kuijpers, A., and
Vroon, P. Z.: Ocean circulation and iceberg discharge in the glacial North Atlantic: Inferences from
unmixing of sediment size distributions, Geology, 30, 555-558, 2002.
Prokopenko, A., Bezrukova, E., Khursevich, G., Solotchina, E., Kuzmin, M., and Tarasov, P.: Climate in
continental interior Asia during the longest interglacial of the past 500 000 years: the new MIS 11
records from Lake Baikal, SE Siberia, Climate of the Past, 6, 31-48, 2010.
Rachmayani, R., Prange, M., Lunt, D. J., Stone, E. J., and Schulz, M.: Sensitivity of the Greenland Ice
Sheet to Interglacial Climate Forcing: MIS 5e Versus MIS 11, Paleoceanography, 32, 1089-1101,
doi:10.1002/2017PA003149, 2017.
Rahmstorf, S.: Bifurcations of the Atlantic thermohaline circulation in response to changes in the
hydrological cycle, Nature, 378, 145-149, 1995.
Raymo, M. E., and Mitrovica, J. X.: Collapse of polar ice sheets during the stage 11 interglacial,
Nature, 483, 453-456, 2012.
Raynaud, D., Barnola, J.-M., Souchez, R., Lorrain, R., Petit, J.-R., Duval, P., and Lipenkov, V. Y.: The
record for marine isotopic stage 11, Nature, 436, 39-40, 2005.
Rebolledo, L., Sepúlveda, J., Lange, C. B., Pantoja, S., Bertrand, S., Hughen, K., and Figueroa, D.: Late
Holocene marine productivity changes in Northern Patagonia-Chile inferred from a multi-proxy
analysis of Jacaf channel sediments, Estuarine, Coastal and Shelf Science, 80, 314-322, 2008.
Reille, M., Beaulieu, J. L. D., Svobodova, H., Andrieu-Ponel, V., and Goeury, C.: Pollen analytical
biostratigraphy of the last five climatic cycles from a long continental sequence from the Velay
region (Massif Central, France), Journal of Quaternary Science: Published for the Quaternary
Research Association, 15, 665-685, 2000.
Reyes, A. V., Carlson, A. E., Beard, B. L., Hatfield, R. G., Stoner, J. S., Winsor, K., Welke, B., and
Ullman, D. J.: South Greenland ice-sheet collapse during marine isotope stage 11, Nature, 510, 525,
1139 2014.

Risebrobakken, B., Dokken, T., Smedsrud, L. H., Andersson, C., Jansen, E., Moros, M., and Ivanova, E.
V.: Early Holocene temperature variability in the Nordic Seas: The role of oceanic heat advection
versus changes in orbital forcing, Paleoceanography, 26, 2011.
Riveiros, N. V., Waelbroeck, C., Skinner, L., Duplessy, J.-C., McManus, J. F., Kandiano, E. S., and
Bauch, H. A.: The "MIS 11 paradox" and ocean circulation: Role of millennial scale events, Earth and
Planetary Science Letters, 371, 258-268, 2013.
Robinson, A., Calov, R., and Ganopolski, A.: Multistability and critical thresholds of the Greenland ice
sheet, Nature Climate Change, 2, 429-432, 2012.



Robinson, A., Alvarez-Solas, J., Calov, R., Ganopolski, A., and Montoya, M.: MIS-11 duration key to
disappearance of the Greenland ice sheet, Nature communications, 8, 16008, 2017a.
Robinson, A., Alvarez-Solas, J., Calov, R., Ganopolski, A., and Montoya, M.: MIS-11 duration key to
disappearance of the Greenland ice sheet, Nature communications, 8, 1-7, 2017b.
Robinson, S. G., and McCave, I. N.: Orbital forcing of bottom-current enhanced sedimentation on
Feni Drift, NE Atlantic, during the mid-Pleistocene, Paleoceanography, 9, 943-972, 1994.
Rothwell, R. G.: Micro-XRF studies of sediment cores: a perspective on capability and application in
the environmental sciences, in: Micro-XRF studies of sediment cores, Springer, 1-21, 2015.
Ruddiman, W. F.: Late Quaternary deposition of ice-rafted sand in the subpolar North Atlantic (lat
40° to 65°N), Geol. Soc. Am. Bull., 88, 1813-1827, 1977.
Ruddiman, W. F.: Cold climate during the closest stage 11 analog to recent millennia, Quaternary
Science Reviews, 24, 1111-1121, 2005.
Sandø, A. B., Nilsen, J. E. Ø., Eldevik, T., and Bentsen, M.: Mechanisms for variable North Atlantic–
Nordic seas exchanges, Journal of Geophysical Research: Oceans, 117,
https://doi.org/10.1029/2012JC008177, 2012.
Sgubin, G., Swingedouw, D., Drijfhout, S., Mary, Y., and Bennabi, A.: Abrupt cooling over the North
Atlantic in modern climate models, Nature Communications, 8, 1-12, 2017a.
Sgubin, G., Swingedouw, D., Drijfhout, S., Mary, Y., and Bennabi, A.: Abrupt cooling over the North
Atlantic in modern climate models, Nature Communications, 8, 2017b.
SHICHI, K., TAKAHARA, H., and KAWAMURO, K.: Vegetation and climate changes during MIS 11 in
southeastern Siberia based on pollen records from Lake Baikal sediment, Japanese Journal of
Palynology, 55, 3-14, 2009.
Siccha, M., and Kucera, M.: ForCenS, a curated database of planktonic foraminifera census counts in
marine surface sediment samples, Scientific data, 4, 170109, 2017.
Smeed, D. A., McCarthy, G. D., Cunningham, S. A., Frajka-Williams, E., Rayner, D., Johns, W. E.,
Meinen, C. S., Baringer, M. O., Moat, B. I., and Duchez, A.: Observed decline of the Atlantic
meridional overturning circulation 2004–2012, Ocean Science, 10, 29-38, 2014.
Smeed, D. A., Josey, S., Beaulieu, C., Johns, W. E., Moat, B. I., Frajka-Williams, E., Rayner, D., Meinen,
C. S., Baringer, M. O., and Bryden, H. L.: The North Atlantic Ocean is in a state of reduced
overturning, Geophysical Research Letters, 45, 1527-1533, 2018.
Solignac, S., Seidenkrantz, M.-S., Jessen, C., Kuijpers, A., Gunvald, A. K., and Olsen, J.: Late-Holocene
sea-surface conditions offshore Newfoundland based on dinoflagellate cysts, The Holocene, 21, 539-
1180   552, 2011.

Stommel, H.: Thermohaline convection with two stable regimes of flow, Tellus, 13, 224-230, 1961.
Stouffer, R. J., Yin, J., Gregory, J., Dixon, K., Spelman, M., Hurlin, W., Weaver, A., Eby, M., Flato, G.,
and Hasumi, H.: Investigating the causes of the response of the thermohaline circulation to past and
future climate changes, Journal of Climate, 19, 1365-1387, 2006.



Sutton, R., and Allen, M. R.: Decadal predictability of North Atlantic sea surface temperature and
climate, Nature, 388, 563-567, 1997.
Swingedouw, D., Rodehacke, C. B., Behrens, E., Menary, M., Olsen, S. M., Gao, Y., Mikolajewicz, U.,
Mignot, J., and Biastoch, A.: Decadal fingerprints of freshwater discharge around Greenland in a
multi-model ensemble, Climate Dynamics, 41, 695-720, 2013.
Tanaka, T., Togashi, S., Kamioka, H., Amakawa, H., Kagami, H., Hamamoto, T., Yuhara, M., Orihashi,
Y., Yoneda, S., and Shimizu, H.: JNdi-1: a neodymium isotopic reference in consistency with LaJolla
neodymium, Chemical Geology, 168, 279-281, 2000.
Team, R. C.: R: A Language and Environment for Statistical Computing (Version 3.5. 2, R Foundation
for Statistical Computing, Vienna, Austria, 2018), There is no corresponding record for this
reference.[Google Scholar], 2019.
Tedesco, M., and Fettweis, X.: 21st century projections of surface mass balance changes for major
drainage systems of the Greenland ice sheet, Environmental Research Letters, 7, 045405, 2012.
Telesiński, M. M., Łącka, M., Kujawa, A., and Zajączkowski, M.: The significance of Atlantic Water
routing in the Nordic Seas: the Holocene perspective, The Holocene, 32, 1104-1116, 2022.
Telford, R. J., Heegaard, E., and Birks, H. J. B.: All age–depth models are wrong: but how badly?,
Quaternary science reviews, 23, 1-5, 2004.
Thibodeau, B., Bauch, H. A., and Pedersen, T. F.: Stratification-induced variations in nutrient
utilization in the Polar North Atlantic during past interglacials, Earth and Planetary Science Letters,
1204    457, 127-135, 2017.

Thornalley, D. J., Oppo, D. W., Ortega, P., Robson, J. I., Brierley, C. M., Davis, R., Hall, I. R., Moffa-
Sanchez, P., Rose, N. L., and Spooner, P. T.: Anomalously weak Labrador Sea convection and Atlantic
overturning during the past 150 years, Nature, 556, 227, 2018.
Tibshirani, R. J., and Efron, B.: An introduction to the bootstrap, Monographs on statistics and
applied probability, 57, 1-436, 1993.
Trachsel, M., and Telford, R. J.: All age–depth models are wrong, but are getting better, The
Holocene, 27, 860-869, 2017.
Tzedakis, P., Andrieu, V., De Beaulieu, J.-L., Crowhurst, S. d., Follieri, M., Hooghiemstra, H., Magri, D.,
Reille, M., Sadori, L., and Shackleton, N.: Comparison of terrestrial and marine records of changing
climate of the last 500,000 years, Earth and Planetary Science Letters, 150, 171-176, 1997.
Tzedakis, P., Hooghiemstra, H., and Pälike, H.: The last 1.35 million years at Tenaghi Philippon:
revised chronostratigraphy and long-term vegetation trends, Quaternary Science Reviews, 25, 3416-
1217    3430, 2006.

Tzedakis, P.: The MIS 11–MIS 1 analogy, southern European vegetation, atmospheric methane and
the" early anthropogenic hypothesis", Clim Past, 6, 131-144, 2010.
Tzedakis, P., Wolff, E., Skinner, L., Brovkin, V., Hodell, D., McManus, J. F., and Raynaud, D.: Can we
predict the duration of an interglacial?, 2012.
Van Rooij, D., Blamart, D., Richter, T., Wheeler, A., Kozachenko, M., and Henriet, J.-P.: Quaternary
sediment dynamics in the Belgica mound province, Porcupine Seabight: ice-rafting events and
contour current processes, International Journal of Earth Sciences, 96, 121, 2007.
Vellinga, M., Dickson, B., and Curry, R.: The changing view on how freshwater impacts the Atlantic
Meridional Overturning Circulation, in: Arctic–subarctic ocean fluxes: Defining the role of the
northern seas in climate, Springer, 289-313, 2008.
Waelbroeck, C., Labeyrie, L., Duplessy, J. C., Guiot, J., Labracherie, M., Leclaire, H., and Duprat, J.:
Improving past sea surface temperature estimates based on planktonic fossil faunas,
Paleoceanography, 13, 272-283, 1998.
Weltje, G. J.: End-member modeling of compositional data: Numerical-statistical algorithms for
solving the explicit mixing problem, Mathematical Geology, 29, 503-549, 1997.
Wood, R. A., Keen, A. B., Mitchell, F. B., and Gregory, J. M.: Changing spatial structure of the
thermohaline circulation on response to atmospheric $CO_2$ forcing in a climate model, Nature, 399,
1235 572-575, 1999.

Worthington, E. L., Moat, B. I., Smeed, D. A., Mecking, J. V., Marsh, R., and McCarthy, G. D.: A 30-
year reconstruction of the Atlantic meridional overturning circulation shows no decline, Ocean Sci.,
17, 285-299, 10.5194/os-17-285-2021, 2021.
Wu, Q., Colin, C., Liu, Z., Douville, E., Dubois-Dauphin, Q., and Frank, N.: New insights into
hydrological exchange between the South China Sea and the Western Pacific Ocean based on the Nd
isotopic composition of seawater, Deep Sea Research Part II: Topical Studies in Oceanography, 122,
1242 25-40, 2015.

Yin, Q., and Berger, A.: Interglacial analogues of the Holocene and its natural near future, Quaternary
Science Reviews, 120, 28-46, 2015.
Yu, L., Gao, Y., and Otterå, O. H.: The sensitivity of the Atlantic meridional overturning circulation to
enhanced freshwater discharge along the entire, eastern and western coast of Greenland, Climate
Dynamics, 46, 1351-1369, 2016.