# Peer review of "Title: Nordic Seas Deep-Water susceptible to enhanced freshwater export to the subpolar"

_Climate of the Past, 2023_

## Author Comment (AC1)

Dear Reviewer 3,

We would like to thank you for helpful comments on our manuscript. Here we have addressed each of the comments and questions in the following format: Each question or comment is re-stated as in the original review of the manuscript in black font. Our response to each comment/question is indented and written in blue 'Calibri font'. All changes made in the manuscript can be found in the TRACK_CHANGES version of the manuscript.

*We noticed that the general comments below are repeated with more detail under **Detailed Comments.** To avoid repetition, we therefore responded to the comments in the Detailed Comment section.*
* * *
**Comments from Reviewer 3**

Review report to "Nordic Seas Deep-Water susceptible to enhanced freshwater export to the subpolar North Atlantic during peak MIS 11" submitted to Climate of the Past by Curran et al.

The manuscript investigates the causes and consequences of a meltwater event in the North Atlantic during MIS11. The authors created new data records of planktic foraminiferal assemblages, XRF, particle size distributions, authigenic epsilon Nd, and benthic foraminifera stable isotopes across the event from Site DSDP 610 at the southern tip of Rockall Bank in the Northeast Atlantic.

The authors interpret the proxy changes they observe to show a deep water current slowing and reduced presence of WTOW at the core site following the freshwater event, and interpret decreased benthic stable carbon isotope data as an increased presence of southern ocean waters. Notably, they interpret differences in the surface water proxies such as reconstructed temperatures and faunal assemblages across the North Atlantic region to indicate that fresh surface waters propagated from the western Nordic Seas into the western North Atlantic and then to the East. They reconcile their findings by assessing that North Atlantic Subpolar Gyre Dynamics are essential for the AMOC behaviour and the freshwater only affected the AMOC by suppressing Nordic Seas deep water formation once it reached the North Atlantic Current entering the Nordic Seas.

I think the manuscript investigates a very important topic that is a good example of how paleoceanography can inform us about climate dynamics and potential future climate evolution. The study site is well suited is in an interesting location and presumably well suited for proxy reconstructions, and the proxy observations appear of high quality. Generally, most observations fit the overall picture of proxy records across the region, supporting their viability. I have a few open questions and concerns about the interpretations, detailed below. While the writing is generally clear, I found it not easy to follow the interpretations, mainly because 1) the figures are not easy to read and 2) many findings from other studies are mentioned but not shown (which, of course, is not always possible). I suggest to improve on these two points wherever possible and added some ideas about the figures at the end.

My main concerns are:

1) The interpretations about changes in the deep circulation. I do not necessarily object the interpretations, but they could be better supported. Several points are listed in the detailed comments below.

2) The sequence of events occurring at the site currently seems insufficiently supported and visualised. An important argument the authors raise is that deep ocean changes occurred before changes at the surface. I am not yet convinced this delay is significant. Maybe showing the records vs. sediment depth or detailing sediment depths, ages, and proxy record changes together with uncertainties could help?

3) The figures could be improved and restructured to support and guide the interpretations better. See also below.

4) The precision of the numbers given appears often too high to be justified and higher than necessary, and generally inconsistent.

**Detailed comments:**

l. 120ff.: How certain is it that the site is bathed mainly and directly by WTOW, and not e.g. by Lower Deep Water (LDW) or North East Atlantic Deep Water (NEADW)? In their Fig. 1b, Crocker et al. (2016) show a section of Rockall Trough with different water masses indicated, which would suggest that Site 610 lies deeper than WTOW today. While it may still have been pathed by northern sourced waters, a dominance of NEADW might make the discussion of the observed deep water changes more complex.

> We acknowledge that modern observations place NEADW at 2417m in the Rockall Trough and rewrote the hydrographic setting accordingly. We also agree that modern WTOW is intermittent on annual timescales and that consequently the variability in the depth range of deep WTOW may not be fully defined for the modern. However, previous studies have shown that the distinct Nd signature of NSOW (e.g., ~-10) has continuously been present in the Rockall Trough (Feni Ridge) at depth deeper than 2000m for the past 44ka (e.g., Site 980 at 2200m; Crocket et al. 2011, Crocket et al. 2016). Especially, the study of Crocket et al. 2016 has specifically addressed the discrepancy between modern observations (e.g., intermittent NSOW) and paleo observations using a comprehensive multi-proxy approach including Nd, B/Ca, 13C and 18O to demonstrate that Nordic Seas Overflow waters were present and significant along the Feni Ridge at depth and timescales relevant to this study.

> Like Crocket et al. 2011 and Crocket et al. 2016, our dataset provides evidence for the presence of NSOW at 610B during MIS11 based on Nd, 13C, and 18O data. We feel that we cannot ignore this evidence, and therefore we cannot ignore that the grainsize data and inferred current flow speeds also incorporate a Overflow Signal.

> We clarified the modern hydrographic setting, specifically, that it differs from paleo-observations in the revised manuscript. We also acknowledge the contribution of deeper water masses including NEADW and AABW in building the Feni Drift.

> Furthermore, we propose to refrain using the water mass name "WTOW" and instead refer to a contribution of NSOW to the overall signal.

l. 245: Is Wu et al. 2015 the best citation for the demonstration of the viability of foraminifera-bound Nd as paleo-circulation tracer and the methods? One well cited (review) paper about this is Tachikawa et al. (2014), so why not cite them, or an earlier paper incorporated there? Also, does the method follow the Tachikawa review? If not, it might be interesting to point out significant differences.

> We have used the analytical method describe in detail by Wu et al., (2015). We have then used this reference in the corrected manuscript. The method used by Wu et al. (2015) is largely accepted in the scientific community and agrees with results obtained by Tachikawa et al. (2014). Tachikawa et al. (2014) present an overview of methodological progress including that of bulk foraminifera and microanalyses within foraminiferal tests. They have demonstrated that Nd-rich phases associated with foraminifera are adhesive nano-scale particles of Mn and Fe oxides and hydroxides, and Mn-rich carbonates formed within layers of foraminiferal calcite. They have then confirmed that Nd isotopic signatures of planktonic foraminifera correspond to bottom water values rather than surface water ones because.

l. 453 and rest of the MS: check consistency of epsilon Nd notation (i.e. the epsilon font family and Nd as subscript or not)

> revised

l. 488 ff.: I cannot find evidence of SST at Site 1305 of 10°C in Fig. 7. If I interpret the figure correct, then SST ranged between ~ 4 and 8 °C. Similarly, the described drop in temperatures is not obvious. Furthermore, in the face of those temperature uncertainties, I suggest to round all given numbers to the full °C, including at many other text paragraphs.

> In the revised figure 7 the SST record from U1305 is more clearly shown. However ` we prefer not to round up temperature estimates units to full units.

l. 507: I understand that the (determination of the) timing of the proxy transitions is limited by the record resolution, but not why this means that the durations are maximum estimates.

> This is because from onset to recovery the event is bracketed by lower NP %. Increased resolution of the time series can therefore only result in a shorter duration of the event.

l. 543: I don't understand what "-38.7 to +3.9m" of sea level equivalent mean. These should be positive numbers, I reckon? Again here, it might be worth to round the numbers.

> These values refer to sea level equivalent of continental Ice relative to modern values. Negative values refer to the potential of lower sea levels and vice versa. In the revised manuscript we clarify this point.

l. 544 f.: I don't understand this sentence. Please clarify how much ice these estimates imply.

> The Greenland Icesheet holds approximately 7m of sea level equivalent in its current size. 39m of sea level equivalent therefore means that there could have been up to 5 Greenland equivalent ice sheets present on earth at 412 ka depending on estimates.

Note that uncertainties are very large and given the lack of terrestrial evidence for so much ice at least in the Northern hemisphere these high-end estimates are very unlikely.

l. 560: Please use consistent precision in numbers, e.g. here the age and its uncertainty.

Here we use the uncertainties as reported in Parker et al. 2023.

l. 566 f.: The IRD concentration numbers could also be rounded.

In the revised manuscript we rounded numbers for IRD to the closest integer.

l. 578: It is not quite clear what anomalous means here. Is it unlike the Holocene? Or colder than the surroundings? Or colder than earlier? Also, should it mean IN the Nordic Seas?

Yes, SSTs are much colder in the Nordic Seas when compared to the Holocene. We revised the sentence to clarify our meaning.

l. 586: "and therefore 412 ka" should be "and therefore date to 412 ka", I think?

Revised as suggested.

l. 596 ff.: Please add references to figures.

Revised as suggested.

l. 605 ff.: The age uncertainties seem very small. Do the authors indeed claim that e.g. the error of 10 years in line 608 is defendable? Maybe adding (or focussing on) the sediment depths of the different discussed signal onsets helps making this discussion more convincing?

The age uncertainties reported here refer to the uncertainties linked to the onset of climate change or ramps that were statistically determined using the Rampfit Function. They do not refer to the absolute errors associated with the age model which is estimated at ± 4 ka. The small error of 10 years highlighted by the reviewer is linked to the high resolution XRF record that shows a very clear and abrupt transition that is statistically estimated to have occurred over 10 year.

l. 611 ff.: Could it be that the changes were brought about by changes in the subsurface to deep ocean and then propagated to the cryosphere and the surface ocean?

It is not quite clear what mechanism the reviewer is referring to.

l. 614 ff.: I think the authors should be very careful when stating Nd isotopic signatures of water masses during MIS11 when they do not cite proof from contemporaneous archives. It is well established that end member Nd isotopic signatures of different water masses have varied through time, often in accordance with long term climatic changes. Hence, citing modern Nd isotopic values for MIS 11 water masses may be interesting in the face of a lack of better information, but must be taken with caution and this should be reflected by the manuscript text. This applies to other parts of the manuscript, too.

In the revised manuscript we state that our interpretation assumes that water mass endmembers during peak MIS 11 are not radically different from today given similar climate boundary conditions.

l. 623 ff.: The reductions in benthic $\delta^{13}C$ have indeed been interpreted as increased presence of SOW before. However, I am wondering about two aspects:

1) Do the carbon and neodymium isotope values fall in the mixing polygon of the different described end members (i.e. WOTW, LSW, LDW)? Do the changes add up to allow for pure changes in the mixing ratios of these water masses? Or are additional processes necessary to explain the evolutions in these two proxies? A cross-plot of these two proxies may already answer these questions.

We have prepared a cross plot for the carbon and neodymium isotope values here. Modern endmember 13C values for water masses are defined as in Repschläger et al. 2015 and Eide et al. 2017 and as in (Dubois-Dauphin et al., 2017;Lambelet et al., 2016) for eNd. The cross plot supports the original interpretation of the manuscript showing that the deepwater proxies are best explained by contributions of Northern sourced overflows and NEADW. We are happy to include the cross plot in the SM of the revised manuscript, including the references used for modern water masses. We also note that 13C are heavier than for the Holocene and Nd are more radiogenic than modern or Holocene values. Both would argue that during MIS11 overflow signal was stronger.

[Figure]

2) Galaasen et al. (2020) also analysed a few samples on their B/Ca ratios, which relate to the seawater carbonate ion concentration. In their Fig. S6 B they show the combined evolution of carbon isotopes and B/Ca across the low-$\delta^{13}C$ events at Site U1305. They interpret "*the association of high (low) C. wuellerstorfi $\delta^{13}C$ with high (low) C. wuellerstorfi B/Ca*" as an

indication of NADW replacement by nutrient rich SOW. However, this association is arguably very weak, with an $R^2 = 0.2$ across their data. I am wondering whether these data do not rather argue against a strong incursion of SOW, but rather other geochemical processes, leading to a (northern sourced?) deep water with high B/Ca and low $\delta^{13}$C ? Maybe the process is similar to those observed by Yu et al. (2008) during the Last Glacial?

We reviewed the discussion in Yu et al. 2008. They propose a decoupling of 13C and CO3 during the LGM that led to lower 13C while CO3 remained high. They argue that during glacial sea-level low stands, exposure of $CaCO_3$-rich continental shelves to atmosphere and physical abrasion of rocks by ice sheets would intensify weathering, which would increase ALK and carbonate ion input to the Nordic Seas and Arctic. Similarly, they argue that the ratio of deep water in the Nordic Seas produced by sea-ice formation induced brine expulsion to that produced by open ocean convection was higher during the LGM than today. This process also increases seawater ALK and $[CO_3^{2-}]$ without any effect on $\delta^{13}$C.

First, the processes described by Yu et al. 2008 invoke long-term glacial processes operating over tens of thousands of years (e.g. weathering) culminating in the LGM while the 412-ka event is much more abrupt and situated in the middle of an interglacial period. Furthermore, at 412ka climate boundary conditions are nothing like the LGM especially at high northern latitudes. We therefore posit that the processes described in Yu et al. 2008 cannot be inferred here.

l. 676 ff.: The authors cite Eldevik et al. (2014) for "since strong deep-water formation in the Nordic Seas requires Atlantic inflow and open-ocean convection". However, in the same sentence they state that "A strong AMOC seems at odds with the western Nordic Seas covered by meltwater". Eldevik et al., however, explicitly state that the relationship of open ocean convection with the THC (or AMOC) is not straight forward, and that "It has in particular been observed that a previously inferred causality (Hansen et al., 2001) between northern deep ventilation and dense overflow from the Nordic Seas does not hold (Olsen et al., 2008)". I think it is important for the whole manuscript to include the premise that this link is not as strong as often assumed. This may be a purely semantic issue.

We removed the second part of the sentence and reference to Eldevik et al. 2014.

Fig. 4: Please describe what the yellow bars indicate in the caption. Could they be used also for Fig. 5 to facilitate a temporal comparison?

Revised as suggested.

Fig. 5: It might be helpful to indicate the color-site specifications in the figure or a legend.

Revised as suggested.

Figs. 5 & 7: It is very hard to see in the figures what is described in the text, and in general to decipher the different records in the figures. Some suggestions on how to improve on this (need to be tested to see their usefulness):

- use the same site-specifi colors (and symbols) for all records from each individual site, and also use these in the map in Fig. 2. It might also be good to make these colors systematic, e.g. red-blue depending on latitude (and another color for Site U1305?), or same/similar colors for same regions.

- mention the general location of each site, e.g. Arctic, Rockall Bank, Reykjanes Ridge, Labrador Sea, somewhere near the sites in the plot or the caption; or alternatively indicate the latitudes.

- same or very similar proxies could be plotted on the same axis

- add legends

- annotate figures in the figure panels

- thicker data lines and possibly larger symbols

- vertical bars to point out certain time periods, such as SST cooling at site 610B.

- indications of proxy uncertainties (as error bars next to the records, not necessarily for each data point)

- focus on showing the data described in the text and omit (or move to supplement) other, apparently less relevant data

We have implemented as many of the above suggestions as we were able to in the revised figures of the manuscript.

Fig. 6: Please indicate that these data are from Site 610 from this study. And should the x-axis not show the age instead of core depth?

Revised as suggested.

All figures with proxy records: I suggest to indicate the event (or its sub-phases) with a vertical bar to ease the interpretations.

Revised as suggested.

Figures: I have the feeling that a figure summarising the most important findings across the region as records would be helpful. For example, it could show records of NPS or SST at different locations in the same panel, then current speed and epsilon Nd at Site 610, and then benthic $\delta^{13}C$ at different sites in another panel all as one stacked record, with the critical time of the meltwater event clearly marked with a vertical bar. The many individual plots showing records may be important to show the diversity of proxy records, but one core figure as a summary would surely help the reader to follow the discussion.

Revised as suggested.

References mentioned:

Crocker, A.J., Chalk, T.B., Bailey, I., Spencer, M.R., Gutjahr, M., Foster, G.L., Wilson, P.A., 2016. Geochemical response of the mid-depth Northeast Atlantic Ocean to freshwater input during Heinrich events 1 to 4. Quaternary Science Reviews 151, 236–254. https://doi.org/10.1016/j.quascirev.2016.08.035

Yu, Jimin, Henry Elderfield, and Alexander M. Piotrowski. "Seawater Carbonate Ion-Δ13C Systematics and Application to Glacial–Interglacial North Atlantic Ocean Circulation." *Earth and Planetary Science Letters* 271, no. 1–4 (July 2008): 209–20. https://doi.org/10.1016/j.epsl.2008.04.010.

Tachikawa, K., Piotrowski, A.M., Bayon, G., 2014. Neodymium associated with foraminiferal carbonate as a recorder of seawater isotopic signatures. Quaternary Science Reviews 88, 1–13. https://doi.org/10.1016/j.quascirev.2013.12.027

---

## Author Comment (AC2)

Dear Reviewer 1,

We would like to thank you for helpful comments on our manuscript. Here we have addressed each of the comments and questions in the following format: Each question or comment is re-stated as in the original review of the manuscript in black font. Our response to each comment/question is indented and written in blue 'Calibri font'. All changes made in the manuscript can be found in the TRACK_CHANGES version of the manuscript.

**Comments from Reviewer 1**

**General Comments.**

Basic premise set out on lines 151-178: The supposition that Feni drift is created and strongly influenced by WTOW is not supported by hydrographic data. Many authors have uncritically repeated the suppositions of Ellett & Roberts (1973), notwithstanding the fact that Dickson and Kidd (1987) had shown that Feni was controlled by Deep Water not the overflow in Rockall Trough. The base of the drift is at ~2500 m in northern Rockall Trough, deepening to >3000 m to the south. WTOW does not affect sediment transport at these depths, a requirement for focussing sediment into a drift on the Rockall margin. Overflow at the Wyville-Thomson Ridge has been supposed by some (e.g. New & Smythe-Wright, 2001) to contribute to deep flow along Feni Ridge, but later work demonstrated that this water (WTOW) mixes so intensely with surface water in its passage over Wyville-Thomson Ridge that it is not dense enough to flow along the bottom below 2200 m, the crest-depth of Feni Ridge at 56° N (which deepens to the south and is at 2417 m for ODP Site 610) (Johnson et al., 2010; 2017). Johnson et al (2017) note that zones of erosion in Northern RT ".... *are seen over a depth range (800–2000 m) coincidental with that of deep WTOW* ..." Below 2000m the recent work of Dubois-Dauphin et al., (2023) using Neodymium isotopes demonstrates that 'deep WTOW' lies above 2000 m while NEADW and LDW occupy the Trough bellow that. So WTOW is not a significant player below 2000 m. If there was a larger amount than present of interglacial freshwater the overflow would have been even less dense.

It is much more likely that Feni is caused by the cyclonic circulation of Lower Deep Water mixed with Southern Source water (traced by silicate concentration) in the Deep Northern Boundary Current (DNBC) of McCartney (1992). WOCE data in Kolterman et al., (2011) show that bottom water (Lower Deep Water, LDW) is about one third SSW farther south at 4500 m. This mixes with overlying Northeast Atlantic Deep Water (NEADW) and enters the cyclonic circulation in Rockall trough along the British Irish margin, exiting around the SE corner of Rockall Bank and Feni Drift (e.g. maps of Knutz et al., 2001, 2007).

Because the authors' data have nothing to do with WTOW, the explanations and discussion must be recast in terms of the history of a more likely water mass, namely NEADW. As this contains some ISOW (which includes NSDW) from the S Iceland Basin (plus SSW), there may be elements of the authors' arguments that remain applicable in a rewritten account. 'WTOW' pervades the Discussion which should be removed and the account recast in terms of more likely water masses.

> We acknowledge that modern observations place NEADW at 2417m in the Rockall Trough and rewrote the hydrographic setting accordingly. We also agree that modern WTOW is intermittent on annual timescales and that consequently the variability in the depth range of deep WTOW may not be fully defined for the

modern. However, previous studies have shown that the distinct Nd signature of NSOW (e.g., ~-10) has continuously been present in the Rockall Trough (Feni Ridge) at depth deeper than 2000m for the past 44ka (e.g., Site 980 at 2200m; Crocket et al. 2011, Crocket et al. 2016). Especially, the study of Crocket et al. 2016 has specifically addressed the discrepancy between modern observations (e.g., intermittent NSOW) and paleo observations using a comprehensive multi-proxy approach including Nd, B/Ca, 13C and 18O to demonstrate that Nordic Seas Overflow waters were present and significant along the Feni Ridge at depth and timescales relevant to this study.

Like Crocket et al. 2011 and Crocket et al. 2016, our dataset provides evidence for the presence of NSOW at 610B during MIS11 based on Nd, 13C, and 18O data. We feel that we cannot ignore this evidence, and therefore we cannot ignore that the grainsize data and inferred current flow speeds also incorporate a Overflow Signal.

We clarified the modern hydrographic setting, specifically, that it differs from paleo-observations in the revised manuscript. We also acknowledge the contribution of deeper water masses including NEADW and AABW in building the Feni Drift.

Furthermore, we propose to refrain using the water mass name "WTOW" and instead refer to a contribution of NSOW to the overall signal.

**Specific Comments**

Intro is OK down to line 116 but needs editing as its too long at ~1500 words. It reads like a partially digested piece of Thesis, and >150 references are not all neccessary.

We have shortened the introduction and hydrographic setting in the revised manuscript.

271-279. Cutting out the sand data above an arbitrary 211 µm on the basis of a paper by soil science workers who found a high coefficient of variation for their sand percentage results is ill advised. The cited work by Polarovski et al. records high CVs for sand which bias the results for the three samples (out of only 13) with sand percentage less than 10%. The reason for removal of the sand is apparently to avoid interference by presence of air bubbles, but if this were a universal problem nobody would ever make measurements of sand with a laser sizer anywhere. This is not the case.

We apologize, we were not clear. The data was not cut at 211um arbitrarily. 211um was chosen because it has the least effect on the total representation of the data while removing all influences of air bubbles. Furthermore, when cut at 211um the data still represents 98.6±1.9 % of the total size fraction analysed.

280-289. An alternative to the end member system for assessing current-controlled sorting of sediment is the plot of sortable silt mean size versus percentage to assess sorting, most recently shown by McCave and Andrews (2019). The EM ratios can be conradictory. McCave and Andrews (2019) pointed out that EMs do not always discriminate well- from poorly-sorted records. Jonkers et al. (2015) proposed that their ratio EM2/EM1 provides a current proxy with no influence of IRD (they say "… *it is possible to correct for the contribution of IRD and obtain an estimate of changes in bottom current speed by using*

*the ratio of EM2/ EM1 ….*"). McCave and Andrews (2019) observe that in the case of very slow current and abundant IRD input, resulting in unsorted fines, this EM ratio is simply a grainsize indicator free of IRD influence, not a speed indicator. The 'mean size ' in the range 7.64 to 66.9 mm (Fig. 6) is not far off the Sortable silt mean size range of 10-63 mm and a cross-plot of this mean size versus the EM ratio could show whether the EM ratio stands up as a flow speed proxy here.

[Figure]

We are happy to provide the cross-plot showing mean SS vs SS%. According to McCave and Andrews (2019) the strong correlation of $r^2$=0.87 in our dataset shows that the sediment record from 610B is well current-sorted and provides a reliable flow history. We include this cross plot as a supplementary figure in the revised manuscript and added the following text in the results: *…we confirmed that sediments at site 610B are indeed current sorted by plotting the sortable silt mean size in the range 7.64 to 66.9 mm against percentage (SS%) in Figure S1 (McCave and Andrews, 2019). The strong correlation of $r^2$=0.87 in our dataset shows that the sediment record from 610B is mostly current-sorted and provides a reliable flow history. We do note however that the sortable silt mean size increases in conjunction with IRD. We therefore perform an endmember analysis to separate the influence of IRD from the current controlled sediments."*

308-313. This is Thesis intro style and not needed here.

The corresponding lines have been removed from the revised manuscript.

318. BP is not appropriate here.

BP has been removed from the revised manuscript.

351-361 inc. table 2. This would be better put into Supplementary material. And do we really need to be told that it is a fortran 77 program ?

We removed this section from the main manuscript and placed them in a separate Supplementary material file.

360 and Table 2. What is the parameter that defines 'WTOW' ? No mention has been made of this previously: If it is the $d^{18}O_{ben}$ then it is not correct as WTOW is not at the bed now (and even less likely in the past if made less dense by meltwater). If it is the EM ratio then the same applies; it is not a record of the flow speed of WTOW, but perhaps of NEADW.

> Table 2 is now in the supplementary material and we refer to the current sorted endmember as a deep water current proxy in the revised manuscript.

381 This 'similarity' is disputable. Fig 4 shows maxima at ~412, 410.5, 409.5 and a broad belt between 407.5 and 406

> We replaced the word similar with consistent.

432-4. Gives justification for line 360 and Table 2, but too late. However, as WTOW is not at the bed here this is an invalid statement. It cannot be a proxy for WTOW. Regretably this also applies to the published paper in CP by Holmes et al (2022).

> As stated above our multi-proxy approach does show that NSOW are present at the core site, especially prior to the event. We changed the wording in the revised manuscript and refer to a contribution of NSOW to the overall signal.

Figure 6. Why is this on a depth axis rather than age as with all the other figures (3-6, 7, 8). It is impossible to correlate information with other figures.

> In the revised manuscript Figure 6 is plotted against age.

546-56. This terrestrial discussion has only a tenuous connection to Hi-lat meltwater.

> We included this discussion to highlight the fact that there is no evidence for large quantities of terrestrial ice that could have caused the observed freshwater signal. We feel that this is important as it separates the event mechanistically from an abrupt collapse such as for example the 8.2 ka event.

579-80. Fram Strait is the gateway into the Nordic Seas from the Arctic so channelling Arctic FW via Fram St would INCREASE its export into the Nordic seas.

> The argument we are trying to make here is that *the opening of a second gateway* (e.g., Canadian Archipelago) could have reduced the pressure on Fram Strait and thereby reduced the overall export via Fram Strait into the Nordic Seas.

603-4. Not NSDW export into RT.

> We reworded this sentence as described above.

607-8. The statement that log (Ti/Ca) data record surface ocean properties is disputable. Obviously it is a record of sediment properties, but given a balance between terrigenous and calcareous (ex surface productivity) components the statement may not be correct. The authors say it is a proxy for variations in lithogenic/biogenic inputs. Change could be entirely lithogenic, i.e. not surface ocean.

> We agree with the reviewer. The Ti/Ca record is closely correlated to IRD not only here but also in Holmes et al. 2022 (e.g. $r^2$=0.73). Since the presence of IRD and the presence of biogenic carbonates are closely linked to surface ocean hydrographic changes we believe that our statement is valid.

618. Dubois-Dauphin 2023 have data at depth of site 610 and deeper but others cited do not. (Citations repeated). See Fig #5 from D-D et al,'23

The reviewer refers to Station MR-2 in Dubois-Dauphin 2023 that reached a maximum depth of 1800m. It is not possible to infer the presence or absence of WTOW below this depth based on this profile.

620. Most authors refer to this as SSW; Southern Source Water.

We changed the acronym as suggested.

The length of the discussion, at nearly 3000 words for a brief excursion in MIS 11, seems grossly too much. Much of the ~900-word section on climate forcing and Ocean atmosphere teleconnections is not really key to the point at hand, analysis of a piece from 415 to 402 ka within the long (~424-374 ka, (LR04)) MIS 11.

The discussion on Climate Forcing, provides a unifying mechanism to processes previously thought at odds with each other. We have shortened the manuscript where possible.

Technical Corrections: typos, etc.A few in Refs:  e.g. 1081, 1167,

We have corrected these as suggested.

References (New)

Dickson, R.R. and Kidd, R.B., 1987. Deep circulation in the southern Rockall Trough - the oceanographic setting of site 610. pp 1061- 1074. *Init. Repts. DSDP*, 94, US Gov. Printing Office, Washington DC,

Dubois-Dauphin, Q., Colin, C., Elliot, M., Förstel, J., Haurine, F., Pinna, R., Douville, E., Frank, N., 2023.  The spatial and temporal distribution of neodymium isotopic composition  within the Rockall Trough. *Progress In Oceanography*, 218, Art# 103127, doi: 10.1016/j.pocean.2023.103127

Ellett, D. and Roberts, D., 1973: The overflow of Norwegian Sea deep water across the Wyville–Thomson Ridge, *Deep-Sea  Res.,* 20, 819–835,

Knutz, P. C.,Austin, W.E.N. and Jones, E.J.W., 2001. Millennial-scale depositional cycles related to British Ice Sheet variability and North Atlantic palaeocirculation since 45 ka B. P., Barra Fan, U.K. margin, *Paleoceanography*, 16, 53– 64.

Knutz, P.C., Zahn, R., Hall, I.R., 2007. Centennial-scale variability of the British Ice Sheet: implications for climate forcing and Atlantic meridional overturning circulation during the last deglaciation. *Paleoceanography* 22, PA1207.

McCartney, M. S. (1992), Recirculation components to the deep boundary current of the northern North Atlantic, *Prog. Oceanogr.*, 29, 283– 383

McCave, I.N., Andrews, J.T., 2019. Distinguishing current effects in sediments delivered to the ocean by ice. I. Principles, methods and examples. *Quat. Sci. Rev*. 212, 92–107.

Koltermann, K.P., Gouretski, V.V., & Jancke, K. (2011). *Hydrographic Atlas of the World Ocean Circulation Experiment (WOCE). Volume 3: Atlantic Ocean* (eds. M. Sparrow et al.), International WOCE Project Office, Southampton, UK, 221 pp.

---

## Author Comment (AC3)

Dear Reviewer 2,

We would like to thank you for helpful comments on our manuscript. Here we have addressed each of the comments and questions in the following format: Each question or comment is re-stated as in the original review of the manuscript in black font. Our response to each comment/question is indented and written in blue 'Calibri font'. All changes made in the manuscript can be found in the TRACK_CHANGES version of the manuscript are highlighted.

*We noticed that the general comments below are repeated with more detail under **Major Science Comments.** To avoid repetition, we therefore responded to the comments in the Major Science Comment section.*
* * *
**Comments from Reviewer 2**

I am an observational oceanographer, so my review focuses on the modern-day oceanography, assumptions made and quality of the manuscript.

This manuscript clearly represents a large body of work. It has two main parts: firstly detailing how the surface properties have changed at the core site, and secondly linking this to possible changes in deep water and then drawing conclusions about the AMOC. I think the first part is fair and do not have any general comments on this. However, linking the observed SST change to deep water and the AMOC is, in my opinion, not clearly shown by the authors and I have concerns about this part of the manuscript.

I am not sure that the assumption that the sedimentary record at Feni Ridge is representative of changes in WTOW, and therefore the AMOC is fair because:

(1) We do not know whether WTOW is the bottom water mass at the Feni Ridge, and the papers the authors cite only shows its presence further north. Logically, I think it must flow south, but I do not know how large an influence it is at the Feni Ridge and whether it is the bottom water mass in contact with the feature. The authors could explore this more.

(2) Two reports not cited by the authors suggest that changes in the Feni Ridge record reflect a lateral redistribution of water masses. I think that the isotope work is interesting and goes some way to possibly indicating that this is not a lateral redistribution but I think this needs to be explored further.

(3) WTOW is the smallest component of the Greenland-Scotland overflow waters.

(4) WTOW flow into the Rockall Trough is very variable. If the changes at the Feni Ridge are due to WTOW, are the changes in WTOW representative of a change in the AMOC strength? Or is it more related to dynamics in the Faroe Shetland Channels changing the amount of overflow water entering the Rockall Trough rather than the Iceland Basin through the Faroe Bank Channel?

My other major concern is that the main finding of the paper and the title hinges on a single finding – that a high-resolution surface record shows a change at $412.29 \pm 0.01$ ka while the lower resolution grain-size analysis (which is attributed to WTOW) shows a reduction at $412.86 \pm 45$ ka. While these are outwith errors, a lower resolution SST record does not show

the same offset. Nor I believe do the foram records. I'm uncertain whether the isotope records ($\varepsilon$Nd and $\delta^{13}$C), which are also used to infer WTOW changes, also show the lag.

Additionally, I am curious whether if the grain size or isotopes were sampled at a similarly high-resolution, the offset between the surface and deep would still occur. I have pretty big concerns that the manuscript premise, title and large sections of the discussion are based on this single finding when others are contradictory.

As well as these scientific issues, I feel that the authors need to do some work to condense certain bits of the manuscript (e.g. the discussion, maybe some of the methodology and introduction) and improve the figures. For example, the captions do not match with the figures and there's a lack of (a), (b) etc labelling of multiple panels making them hard to understand.

**Major science comments**

(1) This manuscript assumes that the sediment at Site 610 is representative of WTOW. I am not sure this is reasonable.

> - Deep WTOW has been observed in the northern and central Rockall Trough hugging the western boundary (e.g. Johnson et al., 2017), but this deep WTOW has not been observed further south than 57.5 N. This is likely because studies have not examined further south than 57.5 N. The WTOW observed at 57.5 N must travel southwards, but the depth at which it is at is unknown.

> - The authors slightly mis-cite the literature e.g. L176. Ellett et al. 1986 and Johnson et al., 2017 shows the presence of deep WTOW in the northern and central Rockall Trough. Neither show the presence of WTOW at the Feni Ridge latitude as suggested.

> - WTOW may not be the water mass in contact with the seabed at Site 610. Data from the southern Rockall Trough show that the deepest water mass originates from Antarctic Bottom Water (e.g. McGrath et al., 2012, New and Smythe-Wright, 2001). This may not be true at the depths of the Feni Ridge but the authors need to look at this further.

New and Smythe-Wright, 2001, Aspects of the circulation in the Rockall Trough, CSR, doi:10.1016/S0278-4343(00)00113-8

McGrath et al., 2012, Chemical characteristics of water masses in the Rockall Trough, DSR, doi: 10.1016/j.dsr.2011.11.007.

> We acknowledge that modern observations place NEADW at 2417m in the Rockall Trough and rewrote the hydrographic setting accordingly. We also agree that modern WTOW is intermittent on annual timescales and that consequently the variability in the depth range of deep WTOW may not be fully defined for the modern. However, previous studies have shown that the distinct Nd signature of NSOW (e.g., ~-10) has continuously been present in the Rockall Trough (Feni Ridge) at depth deeper than 2000m for the past 44ka (e.g., Site 980 at 2200m; Crocket et al. 2011, Crocket et al. 2016). Especially, the study of Crocket et al. 2016 has specifically

addressed the discrepancy between modern observations (e.g., intermittent NSOW) and paleo observations using a comprehensive multi-proxy approach including Nd, B/Ca, 13C and 18O to demonstrate that Nordic Seas Overflow waters were present and significant along the Feni Ridge at depth and timescales relevant to this study.

Like Crocket et al. 2011 and Crocket et al. 2016, our dataset provides evidence for the presence of NSOW at 610B during MIS11 based on Nd, 13C, and 18O data. We feel that we cannot ignore this evidence, and therefore we cannot ignore that the grainsize data and inferred current flow speeds also incorporate a Overflow Signal.

We clarified the modern hydrographic setting, specifically, that it differs from paleo-observations in the revised manuscript. We also acknowledge the contribution of deeper water masses including NEADW and AABW in building the Feni Drift.

Furthermore, we propose to refrain using the water mass name "WTOW" and instead refer to a contribution of NSOW to the overall signal.

(2) The manuscript also assumes that variations in sediment at Site 610 are representative of changes in WTOW strength. Two important missing references (Dickson and Kidd, 1987; Kidd and Hill, 1987) suggest that sedimentary changes at the Feni Ridge appear to be linked to the dominance of southern (i.e. AABW, NADW) origin waters rather than changes in the intensity of NSDW alone. These are reports and may have since been discounted, but I think the authors need to discuss this. Especially as it is fundamental to parts of the manuscript talking about deep water and the AMOC.

Dickson and Kidd, 1987, http://www.deepseadrilling.org/94/volume/dsdp94pt2_36.pdf

Kidd and Hill, 1987, http://www.deepseadrilling.org/94/volume/dsdp94pt2_48.pdf

This may be particularly important because, as the authors state, WTOW is also a variable water mass and well as variability in its flow speed (and therefore transport) there are periods when the water mass is not identifiable (e.g. Johnson et al., 2010).

Johnson et al., 2010, Wyville Thomson Ridge Overflow Water: Spatial and temporal distribution in the Rockall Trough, DSR, doi: 10.1016/j.dsr.2010.07.006

In the revised manuscript we include these two citations and provide a more comprehensive description on the water masses and circulation present at the core site.

As stated above we also provide geochemical evidence for the presence of NSOW at our core site during MIS11 which is also supported by previous paleo observations. It follows that the current flow proxy must therefore record the signal of changing overflows also.

(3) A particular interesting area to me is the isotope work ($\varepsilon$Nd, $\delta^{13}$C, $\delta^{18}$O) which suggests that the sediments show the presence of a northern water mass. I think this part of the manuscript needs to be developed slightly. As you refer to $\delta^{18}$O multiple times I think this should be included on Figure 4.

We included ice volume corrected d18O in Figure 4.

(4) I also have some concerns about how representative WTOW is of variations in the AMOC. WTOW is only a small component of the AMOC lower limb and it is variable in nature (e.g. Sherwin et al., 2008, Østerhus et al., 2019). Changes in WTOW in the Rockall Trough may represent temporal variability in overflow at the ridge (e.g. due to dynamics in the Faroe-Shetland Channels) and a shift in the distribution of overflow water between the Rockall Trough and Iceland Basin (e.g. Stashchuk et al., 2011), rather than changes strength of the lower limb of the AMOC.

Sherwin et al., 2008, Quantifying the overflow across the Wyville Thomson Ridge into the Rockall Trough, DSR, doi: 10.1016/j.dsr.2007.12.006

Stashchuk et al., 2011, Numerical investigation of deep water circulation in the Faroese Channels, DSR, doi: 10.1016/j.dsr.2011.05.005

Østerhus et al., 2019, Arctic Mediterranean exchanges: a consistent volume budget and trends in transports from two decades of observations.

As suggested, we consulted the references provided by the reviewer to evaluate the possibility of a shift in the distribution of overflow water between the Rockall Trough and Iceland Basin.

Overall, observations are limited on longer-term timescales and since observations began WTOW seems to have been intermittent, however, there appears to exist consensus that the interannual variability of WTR overflow varies possibly in concert with the total FBC transport (Sherwin et al. 2008, Hansen et al 2001), which accounts for about one-third of the total overflow. In addition, Stashchuck et al. 2011 propose that the main mechanism that controls the proportion of the outflows into the Iceland Basin and the Wyville Thompson Ridge, is Earth' rotation, further suggesting that WTOW flow is proportional to FBCOW on millennial timescales.

(5) As mentioned above in the 'general comments' I have concerns that the finding that the surface conditions change before the AMOC is based on the relationship between two records when they are of different temporal resolution and the result is not repeated in any other of the records examined.

We agree that higher resolution timeseries of all proxies to match the 0.5cm sample interval of the XRF record would be great. However, at a high-resolution site such as 610B this would have required 250 samples for assemblage counts, Nd, stable isotopes and grain size analysis which was not feasible. It is also important to keep in mind that the different proxy records are measured from the same samples, or in other words from the same depth in the core. This means that all offsets are real and not linked to age model uncertainties. For example, the two surface proxies, Ti/Ca and SST begin to show changes at depths 2963.0 and 2965.0 cm below the seafloor respectively while the two deepwater proxies EM2/EM3 and Nd start to show changes at depth 2973.5 cm or earlier in the case of Nd. In other words, the onset of the deepwater changes precede the observations in the surface records by at least

8.5cm in absolute terms. We have clarified this in the revised version of the manuscript.

(6) The authors define NSDW as 'Nordic Seas Deep Water'. This is a term I've not come across before as in observational oceanography NSDW refers to Norwegian Sea Deep Water.

We corrected as suggested.

(7) At multiple points in the manuscript that authors refer to a 'two-step event'. I find this confusing as the manuscript is focussing on the 412ka event whereas the second step appears to be at ~409ka. I suggest the authors consider changing the wording.

In the revised manuscript we simplify the description of the event.

(8) More generally, I found that the manuscript needs to decide whether to focus purely on the 412ka event or also the 409ka event (or to focus on the wider temporal changes and then narrow down to 412ka). At times I felt it jumped around a little.

In the revised manuscript we simplify the description of the event.

(9) The authors need to make sure to refer to figures/subplots at all appropriate points in the manuscript. This is sometimes missing (e.g. Sections 5.3 and 5.4).

We revised the manuscript accordingly.

**Minor science comments**

(1) L104: This needs rewording. Caesar et al. and Thornally et al. refer to present times while this sentence appears to be relating to MS11.

It is clearly stated that these citations are used to refer to observations "of the recent past". Both datasets place their modern observations in the context of Paleodata using paleo methods going back 1500 years.

(2) L148-149 – while measurements of the AMOC in the North Atlantic began in 2004 (RAPID, with OSNAP post-2014), measurements have been made at the exit of the Labrador Sea in the Deep Western Boundary Current at 53 N since 1997 (e.g. Zantopp et al., 2017) and there have been long measurements of overflows at the Greenland Scotland Ridge (e.g. Østerhus et al., 2019).

Zantopp et al., 2017, From interannual to decadal: 17 years of boundary current transports at the exit of the Labrador Sea, doi:10.1002/2016JC012271

Østerhus et al., 2019, Arctic Mediterranean exchanges: a consistent volume budget and trends in transports from two decades of observations.

This section was cut to streamline the revised manuscript.

(3) L163-164: A more pertinent reference than Johnson et al., 2017 is Sherwin et al., 2008.

Sherwin et al., 2008, Quantifying the overflow across the Wyville Thomson Ridge into the Rockall Trough, DSR, doi: 10.1016/j.dsr.2007.12.006

We replaced Johnson et al with Sherwin et al.

(4) L164-165 – a more up-to-date paper looking at fluxes across the Greenland-Scotland Ridge is Østerhus et al., 2019.

Østerhus et al., 2019, Arctic Mediterranean exchanges: a consistent volume budget and trends in transports from two decades of observations.

We have used this reference as suggested.

(5) L168-169: Holliday ea 2000 and Ellett and Martin, 1973 are not appropriate to reference here as neither investigate whether the Feni Ridge is related to WTOW. I don't think Ellett and Martin, 1973 mention the Feni Ridge – do the authors mean Ellett and Roberts, 1973? Holliday et al., 2020 cite this paper.

Ellett and Roberts, 1973, The overflow of Norwegian Sea Deep Water across the Wyville Thomson Ridge, DSR, doi:10.1016/0011-7471(73)90004-1

This section was rewritten in the revised manuscript.

(6) The authors use WOA98 to reconstruct SST (L213). There's been five releases of WOA since then – why have the authors not used e.g. WOA2018? Does this make any difference?

To the best of our knowledge this has little influence.

(7) L394-395: the wording suggests that G. glutinata is shown on Figure 5 but it isn't.

We corrected the revised manuscript accordingly.

(8) L401-410: I also see a big decrease in NP and the coiling ratio that isn't mentioned.

Yes, the % NP is increasing and so is the coiling ratio. These data are referred to in ll. 412-114 and plotted in figure 5. We have added a reference to figure 5 in the revised manuscript.

(9) Section 6.1: It would aid the reader to briefly say where each core site is (e.g. eastern subpolar North Atlantic, eastern Nordic Seas etc) as well as referring back to Figure 2 (which you do sometimes but not always).

We added a reference to Figure 2 in the revised manuscript.

(10) L848-486, L493-494, L531: To me saying that Site U1305 is downstream of the East Greenland Current implies that it is directly influenced by it - which I don't think it is. From Figure 2 this site appears to be more in the central Labrador Sea whereas the EGC flows down the eastern side of Greenland and then continues as the West Greenland Current flowing up the western side.

Site U1305 is located close to the southwestern extremity of Eirik Drift, off southern Greenland at 57°28.5 N, 48°31.8 W. We reworded the revised manuscript to state that the site is influenced by both EGC and the Irminger Current.

**Comments on Figures**

(1) Figure 4 needs improving

- the different colours mentioned in the figure caption don't exist in the figure

We apologize the colours referred to a previous version of the figure. We have revised the figure caption and removed references to colour

- what are the shaded yellow vertical bars?

We added an explanation to the figure caption: The light green vertical bar marks the onset of the event in the deepwater proxies, while the yellow bar marks the onset of the event in the surface proxies.

- the x-axis should be the same as other figures in the paper (e.g. Fig 6) to enable easy comparison between the two

We revised the x-axis in Figure 6 from depth to age.

(2) I was flicking between Figure 4 and 5 a lot. I think the subplots within the figures need re-organising. Adding SST to Figure 5 would aid the reader as the text compares the SST and foram records. The last two subplots on Figure 4 ($\varepsilon$ND and $\delta^{13}$C) aren't referred to in the text until after Figure 5, the authors maybe better re-ordering the text, or changing how the figures are displayed.

To ease the comparison of the data we have converted Figure 8 into a summary figure.

(3) Figure 5 – the colours referred to in the figure caption again do not match those in the figure. I also suggest the authors mention in the figure caption when y-axes are reversed to aid the reader.

We have made the changes to the figure caption as suggested.

(4) Figure 6 – the figure caption is confusing – it is better to label the subplots and refer to (a), (b) etc. This is especially true if there are two different x-axes (such as on Figure 6).

We have revised the figure as suggested.

- I think the IRD subplot is already shown on Figure 4 (?). If so, does it add anything to repeat it on this figure?

IRD was plotted here again to illustrate two points. First, there is good agreement between Mean Size in the sortable silt fraction and IRD which suggests that mean

size may not be an ideal current proxy in this case. Second, IRD also agrees well with EM1 which represents the IRD endmember. However we removed IRD from the revised figure.

- why do you use depth rather than time as the x-axis on this figure?

We have replotted Figure 6 according to age.

(5) Figure 7 – please can the author check that all subplots within this figure are referred to within the manuscript?

We have reviewed and removed the plots not mentioned in the manuscript.

- the IRD subplot is impossible to read because it is showing too many stations as solid bars. I suggest either using transparent bars, lines, or removing some stations.

We have changed the bar into line plots which improves readability of the data a lot.

- I think this subplot (and the caption) would again benefit from each subplot being labelled (a), (b), (c) etc.

We have revised the figure and caption as suggested.

- it'd be good to use more distinctive colours between the different subplots (if you chose to do this).

We have tried to increase the contrast between colours used to improve readability. We have also included core names next to each line plot which should help with readability.

- please can the authors check that all the subplots are referred to in the manuscript?

We have reviewed this and have removed the plots not mentioned in the manuscript.

(6) I felt I was missing seeing the $\delta^{18}O$ timeseries, could this be added as a subplot to e.g. Figure 4? Or tell the reader in the text (not shown).

We have plotted ice volume corrected 18O in the revised figure 4.

**Technical comments**

(1) L77 – write out $CO_2$ in full first time

Revised.

(2) L105-107: please check this sentence as it didn't make sense to me!

This sentence was revised.

(3) L162-163: …. via **the** Wyville Thomson Ridge…

revised

(4) I thought Table 1 and 2, and maybe Figure 3 could maybe go in the SM as they don't seem integral to the main manuscript to me?

We added Figure 3 to the manuscript because the editor requested it. We removed table 2 to the SM however we prefer to keep Table 1 in the main manuscript as it provides information (Lat, Long, depth) about core sites discussed in the manuscript.

(5) L366: do you mean i.e. rather than e.g.??

Yes, this was corrected in the revised manuscript.

(6) L515-520: this feels out of place to me and possibly not needed.

Here we were highlighting the difference between the Holocene and MIS 11 but have removed this section in an effort to shorten the manuscript.

(7) L533-536: Is a reference needed here?

Foraminifera assemblages have been used to infer the passages of fronts across the SPG into the eastern North Atlantic for other time intervals and these studies were cited in line 533. The sentence in ll 533-536 refers to the 412 event (e.g., this study) and to the best of our knowledge we are the first to infer the passage of fronts for this event based on these data.

(8) L541: define SLE

revised

(9) L617-618: double reference

removed

(10) L624: double 610…

removed

(11) General – you have a lot of acronyms and I think some are unnecessary. They can make it harder for the reader, especially if they are non-standard ones. I recommend going through and removing any that aren't needed.

In the revised manuscript we have reduced the number of acronyms to improve readability.